# Structural basis for the reaction cycle of DASS dicarboxylate transporters

**David B Sauer[1,2], Noah Trebesch[3], Jennifer J Marden[1,2], Nicolette Cocco[1,2], Jinmei Song[1,2], Akiko Koide[4,5], Shohei Koide[4,5,6], Emad Tajkhorshid[3]\*, Da-Neng Wang[1,2]\***

[1]Skirball Institute of Biomolecular Medicine, New York University School of Medicine, New York, United States; [2]Department of Cell Biology, New York University School of Medicine, New York, United States; [3]NIH Center for Macromolecular Modeling and Bioinformatics, Beckman Institute for Advanced Science and Technology, Department of Biochemistry, and Center for Biophysics and Quantitative Biology, University of Illinois at Urbana-Champaign, Urbana, United States; [4]Perlmutter Cancer Center, New York University School of Medicine, New York, United States; [5]Department of Medicine, New York University School of Medicine, New York, United States; [6]Department of Biochemistry and Molecular Pharmacology, New York University School of Medicine, New York, United States

**Abstract** Citrate, α-ketoglutarate and succinate are TCA cycle intermediates that also play essential roles in metabolic signaling and cellular regulation. These di- and tricarboxylates are imported into the cell by the divalent anion sodium symporter (DASS) family of plasma membrane transporters, which contains both cotransporters and exchangers. While DASS proteins transport substrates via an elevator mechanism, to date structures are only available for a single DASS cotransporter protein in a substrate-bound, inward-facing state. We report multiple cryo-EM and X-ray structures in four different states, including three hitherto unseen states, along with molecular dynamics simulations, of both a cotransporter and an exchanger. Comparison of these outward- and inward-facing structures reveal how the transport domain translates and rotates within the framework of the scaffold domain through the transport cycle. Additionally, we propose that DASS transporters ensure substrate coupling by a charge-compensation mechanism, and by structural changes upon substrate release.

**\*For correspondence:**
emad@illinois.edu (ET);
Da-Neng.Wang@med.nyu.edu (D-NW)

**Competing interests:** The authors declare that no competing interests exist.

## Introduction

Citrate and dicarboxylates such as α-ketoglutarate (αKG), succinate and malate are intermediates of the TCA cycle. Furthermore, these molecules play essential roles in metabolic signaling and cellular regulation (*Tannahill et al., 2013*; *Mills et al., 2018*; *Huergo and Dixon, 2015*). In particular, citrate acts as a precursor of fatty acid synthesis and allosterically regulates both fatty acid synthesis and glycolysis. Citrate is also used to synthesize acetyl-CoA for histone acetylation and is therefore essential for the regulation of DNA transcription and replication (*Wellen et al., 2009*). Similarly, cytoplasmic αKG and succinate are important in controlling cell fate. Naive embryonic stem cells that exhibit an elevated αKG-to-succinate ratio maintain pluripotency (*Carey et al., 2015*). In contrast, pancreatic ductal adenocarcinoma cells with p53-deficiency have a lowed αKG-to-succinate ratio, and increasing the cellular concentration of αKG leads to a phenotype similar to that of tumor suppression by p53 restoration (*Morris et al., 2019*).

Mammalian cells import di- and tricarboxylates from the bloodstream via the Na$^+$-dependent citrate transporter (NaCT) and the Na$^+$-dependent dicarboxylate transporters 1 and 3 (NaDC1 and

NaDC3). These plasma membrane proteins belong to the solute carrier 13 (SLC13) gene family (*Markovich and Murer, 2004*; *Bergeron et al., 2013*; *Pajor, 2014*). Loss-of-function mutations in the human NaCT transporter cause a type of encephalopathy (SLC13A5 Deficiency) (*Thevenon et al., 2014*; *Hardies et al., 2015*; *Klotz et al., 2016*). In contrast, knocking out NaCT in mice leads to protection from obesity and insulin resistance, while mutations in the homologous fly gene extends their lifespan (*Birkenfeld et al., 2011*; *Rogina et al., 2000*). Variants in the dicarboxylate transporter NaDC3 cause acute reversible leukoencephalopathy, with accumulation of αKG in the cerebrospinal fluid and urine (*Dewulf et al., 2019*). These central roles of SLC13 proteins in cell metabolism and signaling make them particularly attractive targets for treating obesity, diabetes, cancer and epilepsy (*Huard et al., 2015*; *Pajor et al., 2016*).

The mammalian SLC13 proteins are members of the larger divalent-anion sodium symporter (DASS) family (*Markovich and Murer, 2004*; *Bergeron et al., 2013*; *Pajor, 2014*; *Prakash et al., 2003*). DASS proteins typically have a molecular weight of 45–65 kDa and form obligate homodimers. The majority of DASS proteins are $Na^+$-coupled cotransporters with a transport stoichiometry of one substrate to 2–4 $Na^+$ ions (*Figure 1—figure supplement 1a*). However, other DASS members are exchangers (*Pos et al., 1998*), typically exchanging succinate for other dicarboxylates (*Figure 1—figure supplement 1b*). While sequence analysis strongly suggests a shared fold (*Lolkema and Slotboom, 1998*), these two groups separate into distinct clades on a phylogenetic tree and are thereby named DASS-C and DASS-E for cotransporter/symporter and exchanger/antiporter, respectively (*Figure 1—figure supplement 1c*).

All available DASS structures are of the $Na^+$-driven dicarboxylate transporter VcINDY from *Vibrio cholerae*, in an inward-facing ($C_i$-$Na^+$-S) state (*Figure 1—figure supplement 1d–f*; *Mancusso et al., 2012*; *Nie et al., 2017*). Each protomer consists of a scaffold and a transport domain. In the transport domain, the two carboxylate moieties of the bound substrate are coordinated by two conserved Ser-Asn-Thr (SNT) motifs, which also form part of the $Na^+$-binding sites Na1 and Na2. Structural information, along with cross-linking and computer modeling (*Mancusso et al., 2012*; *Mulligan et al., 2016*), indicates that DASS proteins operate in an elevator-type transport mechanism (*Figure 1a*; *Reyes et al., 2009*; *Drew and Boudker, 2016*; *Garaeva and Slotboom, 2020*). However, without the structure of an outward-facing ($C_o$) conformation structure, a detailed description of the changes between the $C_o$ and $C_i$ conformations for DASS proteins remains lacking. How DASS exchangers translocate the substrates across the membrane is completely unknown.

Furthermore, it is not known how the transporter couples substrate binding and release to the $C_o$ to $C_i$ interconversion (*Stein, 1986*). Studies on multiple DASS-C cotransporters using transport kinetics, electrophysiology and chemical cross-linking have shown that these proteins sequentially bind sodium and then substrate, with this order being reversed during the release process (*Figure 1—figure supplement 1a*; *Wright et al., 1983*; *Yao and Pajor, 2000*; *Hall and Pajor, 2005*; *Pajor et al., 2013*; *Mulligan et al., 2014*). The arrangement of the observed substrate and $Na^+$-sites in the known VcINDY $C_i$-$Na^+$-S state is also consistent with such a mechanism. However, it is unclear how $Na^+$ slippage is avoided, namely, how the transporter undergoes interconversion between the apo $C_i$ and $C_o$ states only, and between the fully loaded $C_i$-$Na^+$-S and $C_o$-$Na^+$-S states, but not between the $Na^+$-only loaded $C_i$-$Na^+$ and $C_o$-$Na^+$ states. This information is essential to understanding the mechanism of SLC13 family disease mutations and the design of drugs to manipulate di- and tricarboxylate import. For the DASS-E exchangers, which we hypothesize follow a typical antiport kinetic mechanism (*Figure 1—figure supplement 1b*), it is even less clear how they translocate the substrates across the membrane.

We aimed to characterize the structural basis of the entire transport cycle of the DASS family using a combination of single particle cryo-EM, X-ray crystallography, molecular dynamics (MD) simulations and transport activity assays.

## Results

### Structure determination of multiple states

The only structurally characterized DASS transporter, VcINDY, has exclusively been observed in a $C_i$-$Na^+$-S state (*Figure 1—figure supplement 1e*; *Mancusso et al., 2012*; *Nie et al., 2017*). To obtain the structure of a DASS protein in a substrate-free, $Na^+$-bound state, we purified VcINDY in 100 mM

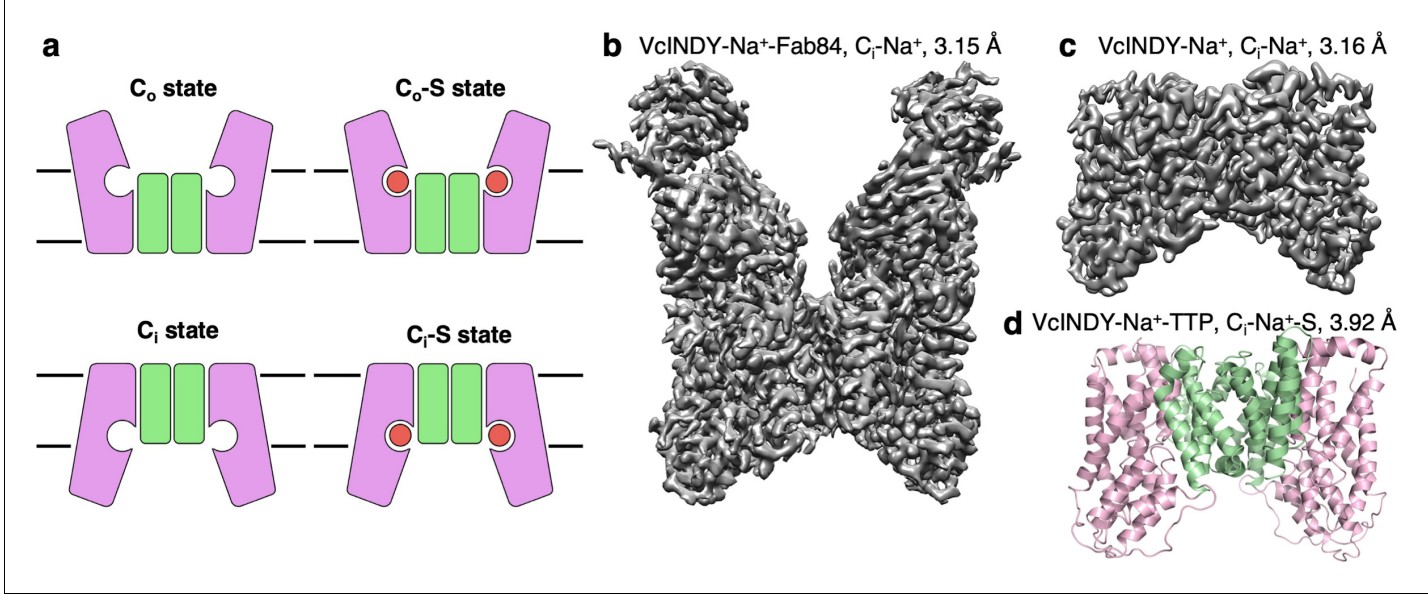

**Figure 1.** Structure determination of the Na$^+$-dependent dicarboxylate cotransporter VcINDY in a C$_i$-Na$^+$ state. (a) The basic conformations and kinetic states of a DASS transporter: outward-facing (C$_o$) and inward-facing (C$_i$) conformations, with or without substrate (S) bound. DASS proteins form a dimer, and each protomer translocates the substrate across the membrane via an elevator-like movement of the transport domain. The scaffold and the transport domains are colored in green and pink, respectively. (b) The 3.15 Å cryo-EM map of VcINDY-Na$^+$-Fab84 complex in nanodiscs, showing the C$_i$-Na$^+$ state. (c) The 3.16 Å cryo-EM map of VcINDY-Na$^+$ in amphipol, showing the C$_i$-Na$^+$ state. (d) The 3.92 Å X-ray structure of VcINDY-Na$^+$-TTP (terephthalate) in detergent, showing the C$_i$-Na$^+$-S state. The scaffold and the transport domains are colored in green and pink, respectively. The online version of this article includes the following figure supplement(s) for figure 1:

**Figure supplement 1.** The DASS family consists of two distinct clades of cotransporters (DASS-C) and exchangers (DASS-E).

**Figure supplement 2.** The VcINDY-Na$^+$ structure determined by cryo-EM is in the C$_i$ state.

Na$^+$, but in the absence of a substrate, for single-particle cryo-electron microscopy (cryo-EM). The total mass of a VcINDY dimer is only 96 kDa, and almost that entire mass is embedded in the membrane. This posed a challenge for single-particle analysis, and therefore we used two separate strategies for cryo-EM sample preparation. The first was to increase the effective particle mass using a synthetic antibody fragment (Fab). Using VcINDY reconstituted in lipid nanodiscs, we screened for Fabs using phage display technology (*Figure 1—figure supplement 2a*; *Fellouse et al., 2007*; *Miller et al., 2012*). In this way we identified Fab84, which was subsequently overexpressed, purified, and mixed with VcINDY in nanodiscs (*Figure 1—figure supplement 2b and c*). The VcINDY-Fab84 structure was determined to 3.15 Å resolution (*Figure 1b*, *Figure 1—figure supplement 2d, e and h*, and *Table 1*). In a parallel approach, we used amphipol polymer (*Huynh et al., 2018*) to preserve the transporter protein (*Figure 1c*, *Figure 1—figure supplement 2f,g & k*, and *Table 1*), which allowed data collection on particularly thin ice to maximize signal-to-noise. This VcINDY structure in amphipol, without Fab, was determined to 3.16 Å resolution. Both maps are at a sufficiently high resolution and quality to allow direct model building, and the structures are nearly identical (r.m.s.d. 0.750 Å). Finally, to obtain clear density of substrate in the binding site of VcINDY, we crystallized the protein in complex with sodium and terephthalate, and solved the X-ray structure to 3.92 Å resolution by molecular replacement (*Figure 1d* and *Table 2*). With two carboxylate moieties at approximately the same distance as those of native substrates, we hypothesized the terephthalate would bind within the VcINDY binding site, while its large benzene ring would provide stronger electron density than succinate, fumarate, or malate.

To obtain the structure of a DASS protein in its C$_o$ conformation, we screened various VcINDY homologs and chose a DASS-E protein from *Lactobacillus acidophilus* (LaINDY) (*Figure 1—figure supplement 1c*). To characterize the function of LaINDY we performed in vivo complementation and transport assays. LaINDY was able to support aerobic growth on αKG as the sole carbon source for an *E. coli* strain in which the only aerobically expressed αKG transporter was knocked out (*Baba et al., 2006*; *Figure 2—figure supplement 1a*). *E. coli* transformed with LaINDY accumulated

**Table 1.** Cryo-EM data collection and structure determination of VcINDY and LaINDY.

| | VcINDY-Na⁺-Fab84 | VcINDY-Na⁺ | LaINDY-apo | LaINDY-Malate | LaINDY-αKG |
|---|---|---|---|---|---|
| EMDB | EMD-21928 | EMD-21904 | EMD-21902 | EMD-21903 | EMD-21905 |
| PDB | 6WW5 | 6WU3 | 6WU1 | 6WU2 | 6WU4 |
| **Data collection** | | | | | |
| Microscope | Arctica-PNCC | Arctica-PNCC | Krios-NYUSoM | Krios-PNCC | Arctica-NYUSoM |
| Magnification | 36,000 | 36,000 | 130,000 | 81,000 | 36,000 |
| Voltage (kV) | 200 | 200 | 300 | 300 | 200 |
| Frames | 826 | 1670 | 3122 | 3255 | 1665 |
| Electron dose (e⁻/Å²) | 44 | 40 | 75.05 | 50 | 46.13 |
| Defocus range (μm) | 0.1–1.5 | 1.0–2.5 | 1.5–2.0 | 0.8–2.5 | 1.5–3.0 |
| Collection mode | Counting | Super-resolution | Counting | Super-resolution | Super-resolution |
| Effective pixel size (Å) | 1.142 | 0.571 | 1.048 | 0.5295 | 0.5575 |
| **Data processing** | | | | | |
| Initial number of particles | 369,769 | 1,072,408 | 3,285,813 | 1,680,542 | 1,564,796 |
| Final number of particles | 92,239 | 192,836 | 278,663 | 277,286 | 64,216 |
| Symmetry imposed | C2 | C2 | C2 | C2 | C2 |
| B-factor sharpening | 119.25 | 136.57 | 140.25 | 176.17 | 36.71 |
| Map resolution* (Å) | 3.15 | 3.16 | 3.09 | 3.36 | 3.71 |
| **Model refinement** | | | | | |
| Non-hydrogen atoms | 13,382 | 6674 | 7708 | 7592 | 7520 |
| Protein residues | 1764 | 890 | 978 | 978 | 978 |
| Ligands | 6 | 0 | 24 | 10 | 0 |
| Mean B factor | | | | | |
| Protein | 28.00 | 75.89 | 10.76 | 11.21 | 95.96 |
| Ligands | 35.87 | - | 61.19 | 58.04 | - |
| RMS deviations | | | | | |
| Bond lengths (Å) | 0.013 | 0.008 | 0.007 | 0.010 | 0.010 |
| Bond angles (°) | 0.947 | 0.602 | 0.605 | 0.689 | 0.677 |
| Molprobity score | 2.41 | 2.58 | 2.08 | 1.84 | 2.14 |
| Clash score | 15.44 | 8.95 | 6.26 | 6.48 | 9.97 |
| Poor rotamers (%) | 1.52 | 6.27 | 2.78 | 0.76 | 0.00 |
| Ramachandran plot | | | | | |
| Favored (%) | 88.87 | 91.65 | 94.15 | 91.99 | 86.86 |
| Allowed (%) | 10.56 | 8.35 | 5.44 | 7.80 | 12.73 |
| Outliers (%) | 0.57 | 0.00 | 0.41 | 0.21 | 0.41 |
| Model Resolution† | 3.5 | 3.5 | 3.3 | 3.6 | 3.7 |

*Resolution determined by Gold-Standard FSC threshold of 0.143 for corrected masked map.

†Resolution determined by FSC threshold of 0.5 for sharpened map.

radioactive succinate from an external solution, likely in exchange for endogenous internal dicarboxylates (*Figure 2—figure supplement 1b*). Subsequent addition of excess external succinate or αKG reduced the amount of internalized radioactive succinate due to LaINDY-facilitated dicarboxylate exchange (*Figure 2a*).

Like other DASS proteins, purified LaINDY formed a dimer in detergent solution (*Figure 2—figure supplement 1d & e*). Observing that the VcINDY structures were nearly identical in amphipol and nanodiscs, we surmised that amphipol was suitable for structure determination of another DASS protein, LaINDY. The LaINDY map obtained in this way was at 3.09 Å resolution, allowing for direct

**Table 2.** X-ray crystallography data collection and structure determination of VcINDY and LaINDY.

|  | VcINDY-TTP | LaINDY-Malate-$\alpha$KG |
|---|---|---|
| PDB | 6WTX | 6WTW |
| Data collection |  |  |
| Space group | P2$_1$ | P2$_1$ |
| Cell dimensions | a = 108.458 Å, b = 103.062 Å, c = 174.446 Å, $\beta$ = 95.848° | a = 91.328 Å, b = 76.609 Å, c = 96.946 Å, $\beta$ = 90.485° |
| Resolution (Å) | 50.0–3.90 | 50.0–2.86 |
| R$_{sym}$(%)* | 11.2 (100.4) | 15.1 (67.2) |
| I/$\sigma$(I) | 13.3 (2.46) | 16.0 (1.48) |
| No. reflections | 145,036 | 192,243 |
| Unique reflections | 33,199 | 30,640 |
| Completeness (%) | 97.4 (98.0) | 99.4 (93.5) |
| Redundancy | 4.4 (4.3) | 6.3 (5.1) |
| CC$_{1/2}$ | 0.962 (0.814) | 0.966 (0.800) |
| **Model refinement** |  |  |
| Resolution (Å) | 3.92 | 2.86 |
| No. reflections | 28,636 | 30,628 |
| R$_{work}$/R$_{free}$ (%)† | 29.0/30.8 | 22.0/27.5 |
| Non-hydrogen atoms | 13,428 | 7504 |
| Protein residues | 1780 | 978 |
| Mean B factor |  |  |
| Protein | 5.64 | 73.46 |
| Ligands | 5.92 | - |
| RMS deviations |  |  |
| Bond lengths (Å) | 0.006 | 0.006 |
| Bond angles (°) | 1.17 | 0.94 |
| Molprobity score | 2.16 | 1.90 |
| Clash score | 9.92 | 9.94 |
| Poor rotamers (%) | 1.42 | 0.0 |
| Ramachandran Plot |  |  |
| Favored (%) | 90.74 | 94.46 |
| Allowed (%) | 6.89 | 4.11 |
| Outliers (%) | 2.37 | 1.44 |

*Values in parentheses are for the highest resolution shell.

†Ten percent of the data were used in the R$_{free}$ calculation.

model building (*Figure 2b*, *Figure 2—figure supplement 1g–i*, and *Table 1*). Notably, the LaINDY structure was in an outward-facing C$_o$ state.

Next, we attempted to determine the structure of LaINDY in its substrate-bound state. To identify substrates suitable for structure determination of such a complex, we examined the thermostability and monodispersity of LaINDY at elevated temperatures in the presence of various potential substrates and substrate analogs (*Figure 2—figure supplement 1f*). Following identification of malate and $\alpha$KG as stabilizers, we determined cryo-EM maps of LaINDY in complex with each to 3.36 Å and 3.71 Å, respectively (*Figure 2c & d*, *Figure 2—figure supplement 1j & k*, and *Table 1*). We also solved a 2.85 Å X-ray structure of LaINDY in the presence of both malate and $\alpha$KG (*Figure 2e* and *Table 2*), using the LaINDY-$\alpha$KG cryo-EM structure as the search model for molecular replacement. As with the apo LaINDY structure, all three LaINDY complex structures were in an outward-facing C$_o$-S state.

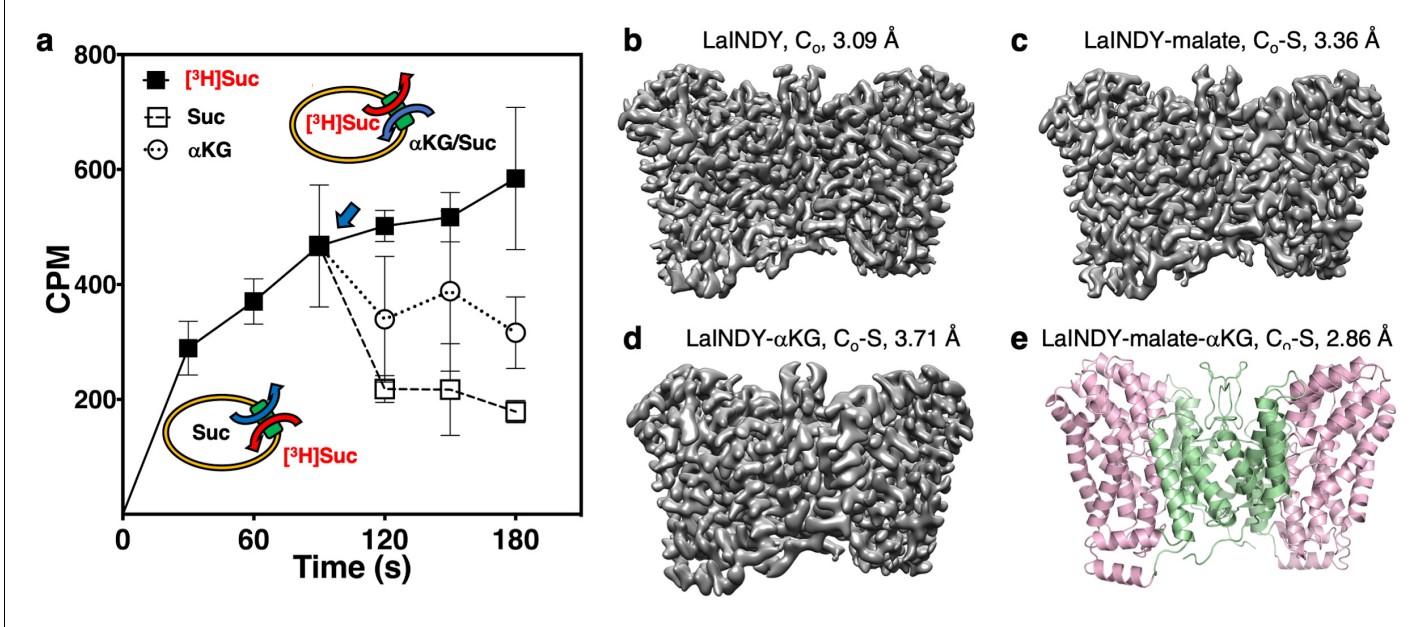

**Figure 2.** Structure determination of the dicarboxylate exchanger LaINDY in $C_o$ and $C_o$-S states. (**a**) Whole-cell transport activity measurements of LaINDY in *E. coli* (N = 3). [³H]succinate was imported into *E. coli* whole cells, driven by the outward gradient of endogenous dicarboxylate such as succinate. When a high concentration of non-radioactive succinate or αKG was added to the external buffer at 90 s (blue arrow), [³H]succinate was exported by LaINDY in exchange for cold succinate or αKG. (**b**) The 3.09 Å cryo-EM map of the dicarboxylate exchanger LaINDY, showing the apo $C_o$ state. (**c**) The 3.36 Å cryo-EM map of LaINDY-malate, showing the $C_o$-S state. (**d**) The 3.71 Å cryo-EM map of LaINDY-αKG, showing the $C_o$-S state. In (**b** – **d**), the cryo-EM samples were prepared in amphipol. (**e**) The 2.86 Å X-ray structure of LaINDY-malate-αKG, showing the $C_o$-S state. The online version of this article includes the following figure supplement(s) for figure 2:

**Figure supplement 1.** LaINDY is tentatively identified as an exchanger, and its structure is in the outward-facing, apo $C_o$ state and substrate-bound $C_o$-S state.

**Figure supplement 2.** Accuracy of structure determination by cryo-EM and X-ray crystallography.

The LaINDY and VcINDY maps were all of sufficient quality to build side chains (*Figure 2—figure supplement 2a–g*), though side-chain rotamers were not always clear at these resolutions. However, we recognized that the significance of comparing structures would be dependent upon the models' accuracy. This accuracy is a concern particularly as two structural methods and multiple instruments were used. We therefore used the models' $C_\alpha$ to C bond lengths as an internal benchmark, as we expect this bond to be insensitive to variations in amino acid sequence, secondary structure, or local environment. The $C_\alpha$ – C bond lengths in our cryo-EM structures, as well as those in the PDB, were typically smaller than the ideal and those determined by X-ray crystallography (*Figure 2—figure supplement 2h–j*). This discrepancy was systematic and independent of map resolution, suggesting it is not the result of microscope calibration errors. Still, the cryo-EM models' accuracy are sufficient to allow us to interpret the observed side chain movements of 1–2 Å.

Finally, to characterize the $C_o$-S to $C_i$-S transition of a DASS protein, we used a custom biased MD simulation protocol (*Figure 6—figure supplement 1a* and Supplementary Note) to drive LaINDY to a $C_i$-S target model suggested by LaINDY's internal inverted repeat topological symmetry (*Video 1* and Supplementary Note). We bookended this induced transition with unbiased MD simulations of LaINDY's $C_o$-S and $C_i$-S states (*Video 2*), which enabled us to characterize LaINDY's equilibrium structural dynamics.

Along with the previous VcINDY structures (*Mancusso et al., 2012*; *Nie et al., 2017*), the newly determined structures of the $C_o$ and $C_i$ conformations in apo and substrate bound states, along with the MD simulations of the $C_o$ to $C_i$ transition, allow us to examine the reaction cycle of DASS transporters. We will begin by describing the outward-facing apo state of LaINDY, and subsequently characterize the structural changes associated with substrate binding to the $C_o$ state, the $C_o$ to $C_i$

transition that carries substrate across the membrane and, finally, substrate release into the cytosol (*Figure 1a*, and *Figure 1—figure supplement 1a and b*).

## LaINDY is in a $C_o$ state

The 3.09 Å cryo-EM map of LaINDY determined in the absence of substrate shows the transporter in its $C_o$ apo state (*Figure 3a and b*). In agreement with LaINDY's apparent mass in detergent solution (*Figure 2—figure supplement 1e*), the map shows a transporter dimer. Each protomer consists of a scaffold domain and a transport domain. The transmembrane topology and domain organization of LaINDY resemble that of VcINDY (*Mancusso et al., 2012*), with the scaffold domain being formed by transmembrane α-helices TMs 1–4 and 7–9, while the transport domain consists of TMs 5, 6, 10 and 11, as well as the helix hairpins $HP_{in}$ and $HP_{out}$ (*Figure 3a-c*). However, for both hairpins, a bend is found in the second helix, at Val163 in $HP_{in}b$ and Ala404 in $HP_{out}b$. Another new structural feature of LaINDY is an extrusion near the dimer interface into the periplasmic space formed by a sequence insertion between TM3 and TM4.

The scaffold and transport domains are linked via two horizontal helices and two loops near the membrane surface, H4c and L4-$HP_{in}$ on the cytosolic side and H9c and L9-$HP_{out}$ on the extracellular side (*Figures 3c* and *4a*). A third linker is formed by a cytoplasmic helix between TM6 and TM7 (H6b), where the equivalent region in VcINDY exists as a long loop (*Figure 4b*). The domain interface is largely formed by branched or short hydrophobic residues, with only two hydrogen bonds. This results in a smooth domain interface, similar to that observed in the elevator transporter $Glt_{Ph}$ (*Reyes et al., 2009*).

While LaINDY's topology is similar to that of VcINDY, the relative domain positions are

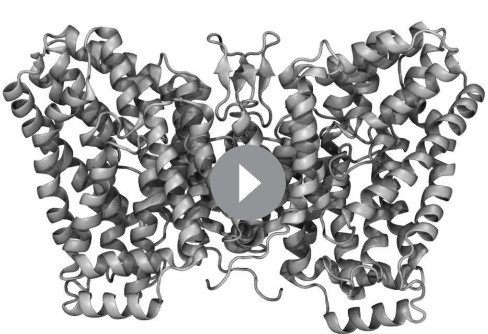

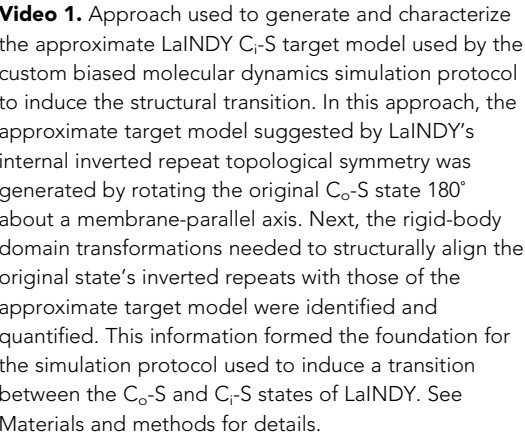

**Video 1.** Approach used to generate and characterize the approximate LaINDY $C_i$-S target model used by the custom biased molecular dynamics simulation protocol to induce the structural transition. In this approach, the approximate target model suggested by LaINDY's internal inverted repeat topological symmetry was generated by rotating the original $C_o$-S state 180° about a membrane-parallel axis. Next, the rigid-body domain transformations needed to structurally align the original state's inverted repeats with those of the approximate target model were identified and quantified. This information formed the foundation for the simulation protocol used to induce a transition between the $C_o$-S and $C_i$-S states of LaINDY. See Materials and methods for details.
https://elifesciences.org/articles/61350#video1

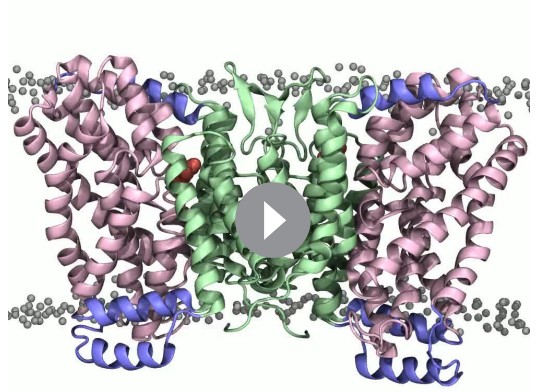

**Video 2.** Molecular dynamics simulation of LaINDY. All stages of the simulation are shown (370 ns), including the unbiased simulation of the $C_o$-S state (100 ns), the induced transition to the approximate $C_i$-S target (100 ns), and the unbiased simulation of the $C_i$-S state (100 ns). Transport domains are shown in light pink; scaffold domain in green; helices H4c, H6b, and H9c in purple; bound succinate in dark pink; and lipid phosphorous atoms in gray. Structural alignment was performed using the scaffold domain's center of mass along the membrane-normal axis and using the transport domains' centers of mass along the membrane-parallel axes.
https://elifesciences.org/articles/61350#video2

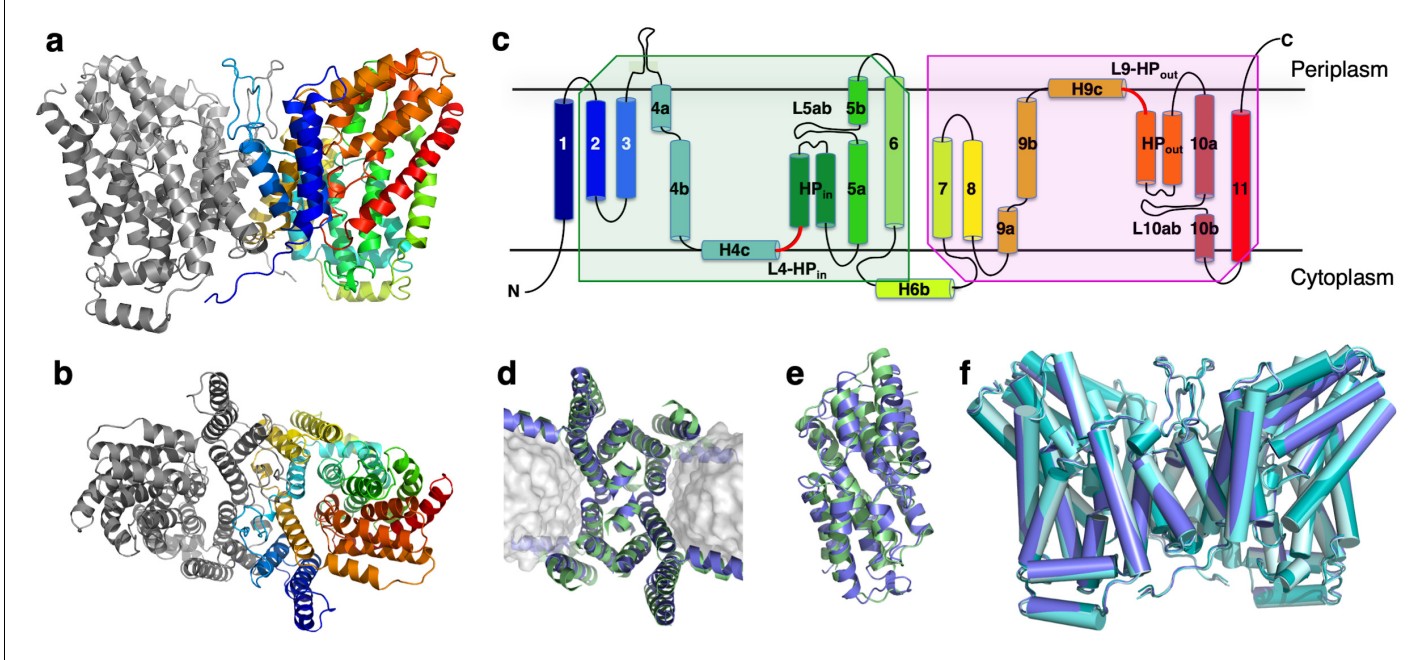

**Figure 3.** Structure of LaINDY and its structural homology to VcINDY. The 3.09 Å C_o structure of apo LaINDY dimer determined by cryo-EM as viewed from (**a**) within the membrane plane and (**b**) the periplasm. (**c**) Topology of LaINDY. Unique to LaINDY is an extrusion near the dimer interface into the periplasmic space, formed by a sequence insertion between TM3 and TM4. The two hinge loop regions between the scaffold and the transport domain, L4-HP_in and L9-HP_out, are colored red. Structural alignment of LaINDY (blue) and VcINDY (PDB ID: 5UL9, green) between (**d**) the scaffold domains and (**e**) the transport domains. (**f**) Overlay of the LaINDY-apo structure (blue) with its three substrate-bound structures, LaINDY-malate cryo-EM structure (pale blue), LaINDY-αKG cryo-EM structure (aquamarine) and LaINDY-malate-αKG X-ray structure (teal).

different. The transport domain is oriented toward the extracellular side with its substrate binding site facing the periplasm, yielding a $C_o$ conformation (*Figure 3a*). The individual LaINDY and VcINDY domains exhibit strong structural homology, with backbone r.m.s.d.s of 2.897 Å and 2.044 Å for the scaffold and transport domains, respectively (*Figure 3d and e*). Compared with VcINDY, the transport domain in LaINDY is repositioned by 13.0 Å towards the periplasm with a 37.4° rotation (*Figure 4*). The domains' structural conservation and relative positions agree with the notion that DASS proteins operate via a rigid-body, elevator-type movement of the transport domain (*Figure 1a*).

## LaINDY has Na⁺ surrogate side-chains near the substrate binding site

The structures of LaINDY determined in malate, αKG and in the malate/αKG mixture are in an outward-facing, substrate-bound ($C_o$-S) state (*Figure 2e* and *Figure 2—figure supplement 1j & k*). The architecture of the LaINDY binding site is similar to that of VcINDY. Notably, within the binding site, density corresponding to substrate is seen in the X-ray omit map of LaINDY-malate-αKG, and in the LaINDY-αKG cryo-EM difference map, in a similar mode to substrate binding in VcINDY (*Figure 5—figure supplement 1e & f*). All three $C_o$-S structures are not only similar to each other but, importantly, also to the LaINDY $C_o$ apo structure, with pairwise r.m.s.d.s of 0.41–0.52 Å (*Figure 3f*), indicating that substrate binding introduces little conformational change.

In contrast to the similarity at the substrate binding site, major differences between LaINDY and VcINDY are seen at the cation binding sites. At the Na1 site of VcINDY (*Figure 5—figure supplement 1f*), an arginine (Arg159) is found in LaINDY, stabilized by a salt bridge with Glu146 (*Figure 5a* and *Figure 5—figure supplement 1a & c*). Similarly, a histidine (His392) in LaINDY is located at the equivalent of the VcINDY Na2 site, with another histidine (His401) located 4 Å on its extracellular side. Indeed, these residues are conserved in DASS exchangers but absent in cotransporters (*Figure 1—figure supplement 1c*). Furthermore, the cation binding sites are more completely enclosed by the surrounding loops in LaINDY compared to VcINDY, facilitated by DASS exchanger specific insertions in the HP_in and HP_out and L10ab regions (*Figure 1—figure supplement 1d*). Based on

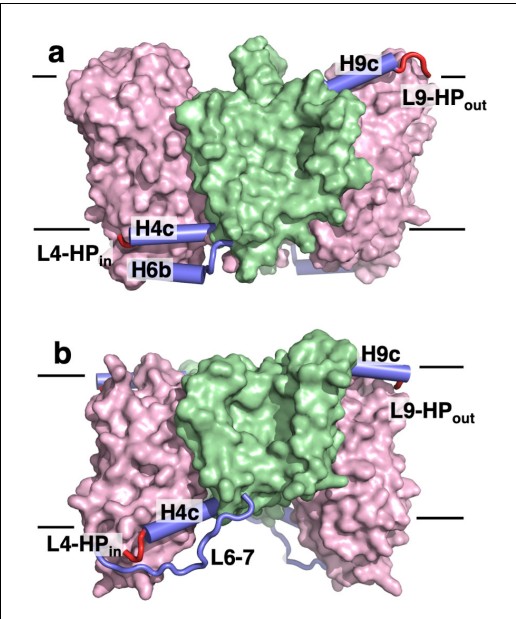

**Figure 4.** The transport domains of DASS proteins are cradled by helical arms. The transport and scaffold domains of (**a**) LaINDY and (**b**) VcINDY are shown in surface presentation, with the arm helices H4c, H6b, and H9c shown as blue cylinders, loop L6-7 as a blue wire, and the connecting loops L4-HPin and L9-HPout as red wires.

these well enclosed and sterically occupied cation binding sites, it is unlikely that sodium ions bind to LaINDY as Na1 and Na2 do in VcINDY. Rather, we hypothesize these two positively-charged residues in exchangers act as permanent surrogates of the $Na^+$ ions in cotransporters. Such $Na^+$ ion substitutions have previously been observed in other transporters, which makes their substrate transport independent of sodium (*Shaffer et al., 2009*; *Kalayil et al., 2013*).

## The transport domain moves as a rigid body within the scaffold domain's arms

The structure determination of LaINDY and VcINDY in multiple states provides an opportunity to characterize conformational changes of a DASS protein during the $C_o$-S to $C_i$-S transition. MD simulations of succinate-bound LaINDY revealed how the transition between its $C_o$-S and $C_i$-S states is realized. The transition consists of a 39° rigid-body rotation and an 8.3 Å translation of the transport domain relative to the scaffold domain (*Figure 6—figure supplement 1b–d* and *Video 2*). The two 'arm' helices, H4c and H9c, are fixed in space with respect to the scaffold domain and cradle the transport domain during the $C_o$-S to $C_i$-S transition (*Figure 6c–e* and *Video 3*). Such arm rigidity agrees with the conserved salt bridge, between Arg122 and Glu283, and bulky-residue interactions at the elbows connecting H4c and H9c to the scaffold domain (*Figure 6a & b*). The importance of the Arg122 to Glu283 salt bridge is also consistent with recent observations in NaCT, where transport activity was abolished by mutations of the equivalent arginine (*Khamaysi et al., 2020*), probably by disrupting the conserved salt bridge between H4c and TM7. In contrast to the rigidity at the elbows, flexibility of the hinge loops L4-HP$_{in}$ and L9-HP$_{out}$ at the other end of the arm helices allows transport domain movement (*Figure 4*). During the $C_o$-S to $C_i$-S transition the angle between the arm helices and the hairpin helices changes by approximately 30° at both hinges (*Figure 6f*), allowing the overall rotation and translation of the transport domain.

Despite the large domain movements of LaINDY during MD simulations of the $C_o$-S to $C_i$-S transition, succinate stably binds without significant changes within the binding site (*Figure 5b*), though the depth of substrate binding correlates with the orientation of Asn156 (*Figure 6—figure supplement 1g,h & I*). Furthermore, the conformations and interactions of the $Na^+$ surrogate side chains, Arg159 and His392, are generally stable through the conformational transition (*Figure 5—figure supplement 1b & d*).

## Substrate release from VcINDY causes significant structural changes

The two cryo-EM structures of VcINDY are in a substrate-free, inward-facing ($C_i$-$Na^+$) state (*Figure 1b & c*) and provide an opportunity to describe the conformational changes that occur upon substrate release. The $C_i$-$Na^+$ VcINDY structures are very similar to each other, with an r.m.s.d of 0.75 Å (*Figure 7—figure supplement 1b*). However, compared with the $C_i$-$Na^+$-S structures, the substrate-free structures display local changes throughout the protein.

At the substrate binding site of both cryo-EM VcINDY structures, Pro422 at the N-terminus of TM10b in the substrate free structures moved by up to 1.5 Å (*Figure 7a*). Concurrently, on the cytoplasmic surface, His432 at the C-terminus of the same helix is rotated by 73° (*Figure 7—figure supplement 1a*). This movement of His432 causes a steric clash with its neighbor, Tyr178 of TM5a,

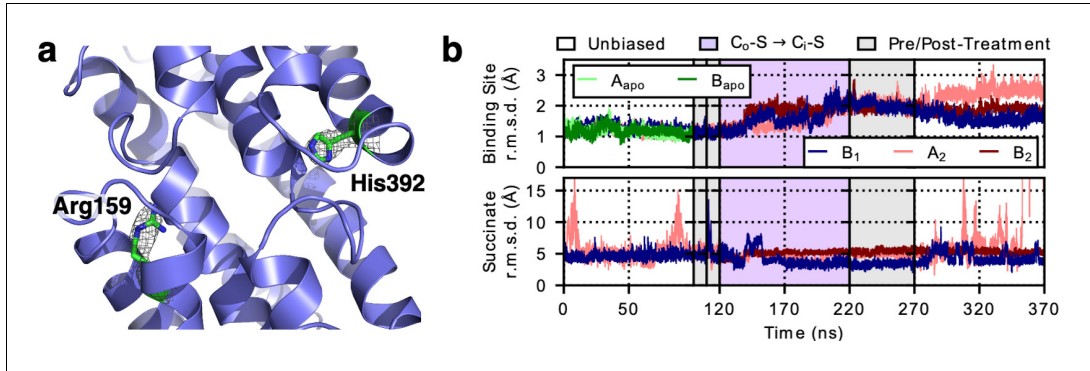

**Figure 5.** LaINDY rigidity when binding substrate. (a) The side chain of Arg159 in LaINDY is found at the location equivalent to the Na1 site in VcINDY, while His392 is found in the Na2 site. Both Arg159 and His392 are conserved in DASS exchangers but absent in cotransporters. These two positively-charged residues in exchangers are hypothesized to act as permanent surrogates of the $Na^+$ ions in cotransporters. (b) Time series of the r.m.s.d. (of $C_\alpha$ and heavy side-chain atoms) of the binding site and the substrate (i.e., succinate) of apo LaINDY (protomers $A_{apo}$ and $B_{apo}$) and substrate-bound LaINDY (protomers $B_1$, $A_2$, and $B_2$). R.m.s.d. values were calculated by comparing the frames of the MD simulations with the X-ray crystal structure after overlaying the helices of the transport domain. The succinate is well ordered during the transition but exhibits increased mobility in the $C_o$ and $C_i$ conformations, corresponding to substrate binding and release.

The online version of this article includes the following figure supplement(s) for figure 5:

**Figure supplement 1.** Charged residues in the binding site of LaINDY act equivalently to $Na^+$ in VcINDY.

inducing a rotation in that side chain by 41°. One helix turn away, the side chain of Arg175 moves closer to Glu437 of TM11, forming a salt bridge. These rearrangements lead to movements of $HP_{in}a$, $HP_{in}b$ and TM5a by 1.4 Å toward the scaffold domain. Supporting the importance of these interactions, human NaCT's transport activity is abolished when the equivalent of Glu437 is mutated to histidine, resulting in SLC13A5 Deficiency (*Hardies et al., 2015*).

Comparison of the previous $C_i$-$Na^+$-S (*Mancusso et al., 2012*; *Nie et al., 2017*) and amphipol preserved $C_i$-$Na^+$ VcINDY structures also revealed prominent changes on the periplasmic surface (*Figure 7b* and *Video 4*). The loop connecting $HP_{out}b$ and TM10a, from Ala395 to Pro400, has moved on the periplasmic surface. The C-terminus of $HP_{out}b$ also unwinds by one turn (Val392 – Glu394) in the amphipol structure. As a result, the conserved salt bridge between Glu394 with Lys337 of H9b from the scaffold domain breaks, and Phe396 moves away from its contact with H9c. Such structural changes agree with observations of human NaCT mutations, at positions equivalent to Pro400 and Val401 in VcINDY, which abolish transport and cause SLC13A5 Deficiency (*Thevenon et al., 2014*; *Klotz et al., 2016*). Finally, the side chain of Trp461 is inserted between TM6, TM10a and TM11, packing against another conserved aromatic residue, Phe220. Notably, when bound to Fab84, the loop connecting $HP_{out}b$ and TM10a in VcINDY is midway between the $C_i$-$Na^+$-S and $C_i$-$Na^+$ structures. This agrees with Fab84's epitope covering the periplasmic surface of VcINDY and binding both apo and substrate-bound states (*Figure 7—figure supplement 1c & d*). Also, a complete lipid molecule from the periplasmic leaflet was found at the interface between the transport and scaffold domains, interacting with TM1, TM2 and $HP_{out}a$, and therefore may be involved in regulating conformational changes (*Figure 7—figure supplement 1e*).

## Discussion

In this work, we report cryo-EM and X-ray structures of two DASS proteins in four states as well as MD simulations of the $C_o$-S to $C_i$-S transition. Of these structures, the $C_i$-$Na^+$, $C_o$, and $C_o$-S are previously un-observed states of the DASS transport cycle. Together with the previously published structures of VcINDY in its $C_i$-$Na^+$-S state, these results give us a much clearer understanding of the structural basis of transport. The gallery of structures allows us to more completely characterize the reaction cycle of DASS transporters. Thereby, we advance the elevator mechanism of the family

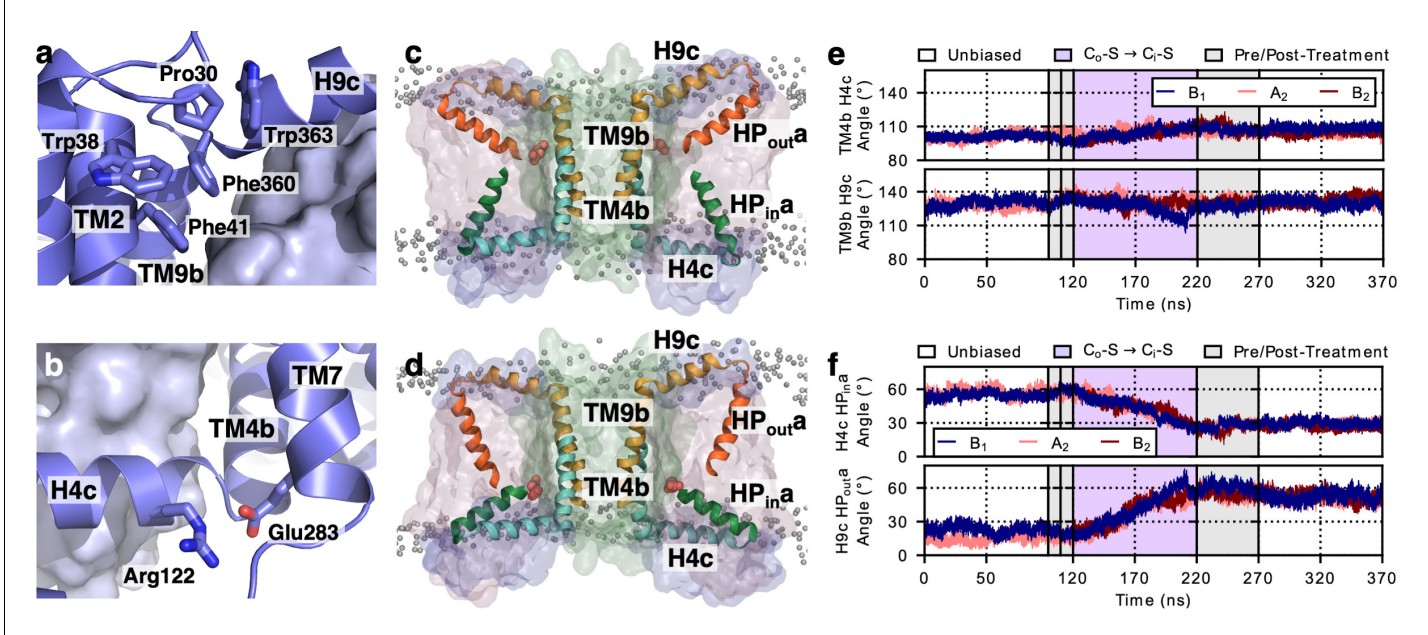

**Figure 6.** Structural changes during the LaINDY $C_o$-S to $C_i$-S transition. (**a**) Bulky residues pack around the elbow preceding the H9c arm of LaINDY. In LaINDY structural interactions among five conserved bulky residues at the junction between the N-terminus of H4c and the core of the scaffold domain make the TM4b – H4c angle rigid. (**b**) A conserved salt bridge is formed between Arg122 of arm H4c and Glu283 of TM7 in the scaffold domain. This salt bridge helps to keep the angle between TM9b and H9c rigid. (**c**) Representative MD structure from the simulation of the LaINDY $C_o$-S state. (**d**) Representative MD structure from the simulation of the LaINDY $C_i$-S state. Between the $C_o$-S and $C_i$-S states, the angles at TM4b – H4c and at TM9b – H9c stay rigid, while the angles at L4-HP$_{in}$ and at L9- HP$_{out}$ change. The change in orientation of HP$_{in}$a to H4c and HP$_{out}$a to H9c accompany the translation and rotation of the transport domain within the framework formed by H4c and H9c and the rest of the scaffold domain. Time series from MD simulations of LaINDY (protomers $B_1$, $A_2$, and $B_2$) showing the structural change at (**e**) the elbow and (**f**) hinge regions. During the $C_o$-S to $C_i$-S transition, while the angles at the two elbows are both rigid, the angle of HP$_{in}$a relative to H4c and HP$_{out}$a relative to H9c changed by 27° and 33°, respectively.

The online version of this article includes the following figure supplement(s) for figure 6:

**Figure supplement 1.** Molecular dynamics simulations of LaINDY.

from a conceptual model into an atomic description of the transport domain's movement within the framework of the scaffold domain.

The structures of LaINDY that we have determined represent the first outward-facing structures of any DASS family protein. These structures generally agree with a previous model of VcINDY in its $C_o$ conformation (*Mulligan et al., 2016*), proposed based on the inverted-topology structural repeat and cross-linking distance constraints, with an r.m.s.d of 3.7 Å for the backbone atoms. As noted, the occupation of the Na1 and Na2 sites by conserved basic residues, and absence of any other apparent sodium densities, suggest LaINDY is a DASS exchanger. This is supported by its phylogeny and ability to catalyze succinate-dicarboxylate exchange. However, further experiments, preferably in reconstituted proteoliposomes, will be needed to examine the sodium and proton dependence of transport, and confirm strict substrate coupling in the exchange reaction.

Comparison of the outward-facing and the inward-facing structures, along with MD simulation results, immediately suggests how a DASS protein operates through an elevator-type movement of the transport domain within each protomer (*Figure 6c & d*). The transport domain moves within the framework formed by the two horizontal α-helix arms on opposing membrane surfaces during the reaction cycle, alternating the substrate binding site between the two sides of the membrane. This mechanism is similar to that proposed for the glutamate transporter Glt$_{Ph}$, although the two arms of Glt$_{Ph}$ are not on the membrane surface but rather transmembrane helices (*Reyes et al., 2009*).

Analyzing the various conformations also enables us to suggest how substrate binding leads to transporter conformational changes while preventing slippage, or unproductive $C_o$ to $C_i$ transitions.

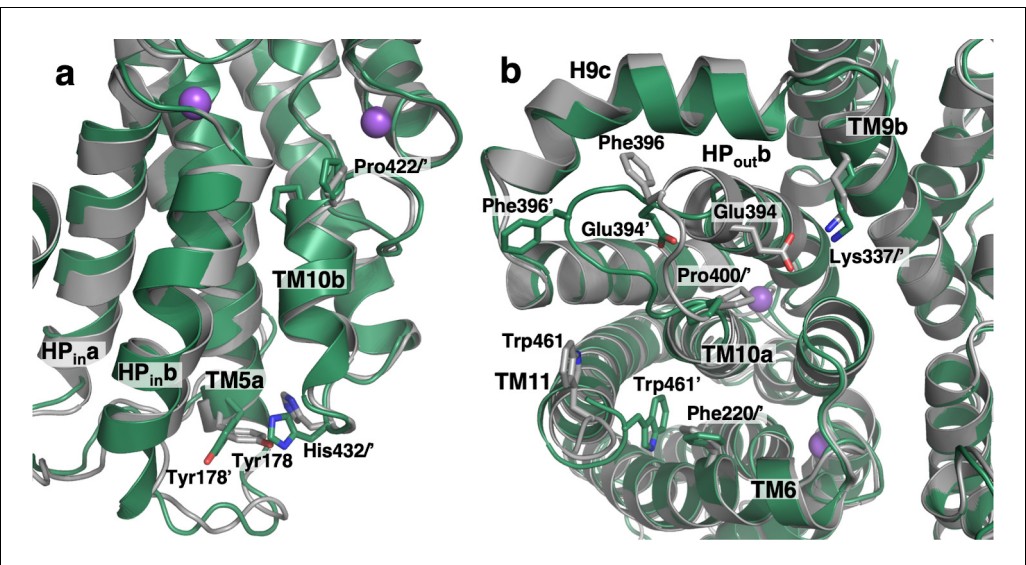

**Figure 7.** Substrate release-induced conformational changes in the $C_i$ state of VcINDY. The cryo-EM structure of VcINDY in a $C_i$-$Na^+$ state, determined in amphipol (dark green), is superimposed on the $C_i$-$Na^+$-S state X-ray structure (grey). Amino acids of the $C_i$-$Na^+$-S and $C_i$-$Na^+$ states are labeled without and with an apostrophe, respectively. (a) At the substrate-binding site Pro422 at the N-terminus of TM10b moves closer to the center in the substrate free structures by 1.5 Å. Concurrently, at the C-terminus of the same helix on the cytoplasmic surface His432 is rotated by 73°. This movement of His432 causes a steric clash with its neighbor, Tyr178, inducing a rotation in that side chain by 41°. These rearrangements lead to movements of $HP_{in}a$, $HP_{in}b$ and TM5a by up to 1.4 Å toward the scaffold domain. (b) On the extracellular surface, the C-terminus of $HP_{out}b$ unwinds by one turn and the entire loop connecting $HP_{out}b$ and TM10a, from Val392 to Pro400, extrudes toward the lateral edge of the protein. As a result, the conserved salt bridge between Glu394 and Lys337 of TM9b breaks, and Phe396 moves away from its contact with H9c. The last four residues at the C-terminus of the protein, Leu459 to Gln462, move closer to the protein surface and the side chain of Trp461 inserts between TM6, TM10a and TM11, packing against another aromatic residue, Phe220.

The online version of this article includes the following figure supplement(s) for figure 7:

**Figure supplement 1.** Lipid and Fab binding of VcINDY, and its conformational changes between $C_i$-$Na^+$-S and $C_i$-$Na^+$ states.

In this regard, DASS cotransporters and exchangers appear to employ both unique and shared mechanisms.

Previous experimental data support that $Na^+$-driven DASS cotransporters operate via an ordered sequence (*Wright et al., 1983*; *Yao and Pajor, 2000*; *Hall and Pajor, 2005*; *Pajor et al., 2013*; *Mulligan et al., 2014*), namely, $Na^+$ binding induces substrate binding, while substrate release precedes $Na^+$ release. For VcINDY, we have now observed that substrate release in the cytoplasm induces conformational changes as the transporter transitions between the $C_i$-$Na^+$-S and $C_i$-$Na^+$ states. Specifically, substrate release leads to an unwinding of the C-terminus of TM11, loop movements, and side chain rotations, resulting in significant changes in local helix packing and protein compactness (*Figure 7*). This is distinguished from $Glt_{Ph}$ and homologs in which one or two hairpin gates directly pack against the scaffold domain and block unproductive conformational changes (*Reyes et al., 2009*; *Garaeva et al., 2019*; *Arkhipova et al., 2020*).

In contrast, the exchanger LaINDY exhibits no major conformational changes between the apo and substrate-bound outward-facing states. As two positively-charged residues are found to occupy the cation binding sites, we propose that the DASS exchangers limit slippage, or unproductive conformational changes, via a charge compensation mechanism. In the apo state, where the binding sites have a net positive charge, the hydrophobic surface of the scaffold domain would be an electrostatic barrier to the movement of the transport domain. Only when divalent substrate binds and the net charge is neutralized can the transporter freely exchange between the $C_i$ and $C_o$ conformations, ensuring a one-to-one stoichiometric exchange of substrates.

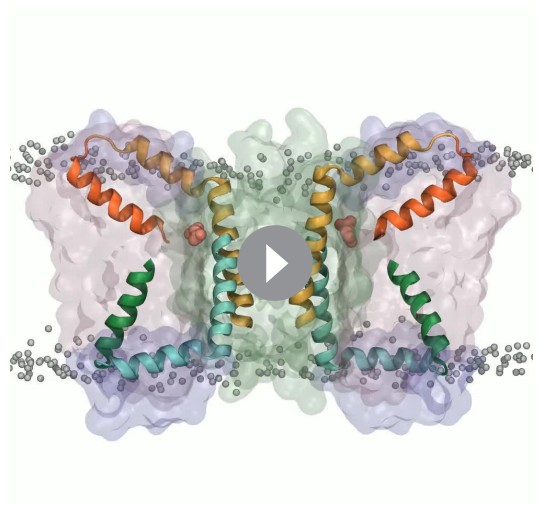

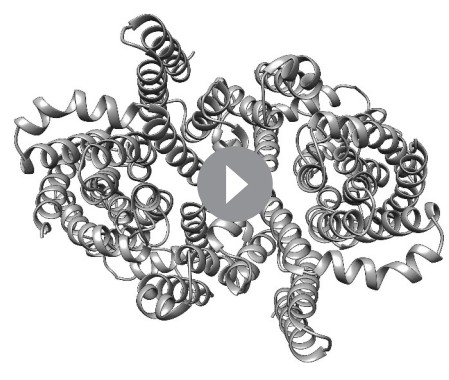

**Video 4.** Linear interpolation of VcINDY between $C_i$-Na$^+$-S and $C_i$-Na$^+$ states. Morph of VcINDY between $C_i$-Na$^+$-S and $C_i$-Na$^+$ states by linear interpolation. https://elifesciences.org/articles/61350#video4

**Video 3.** Structural dynamics of the connections between the transport and scaffold domains during the LaINDY $C_o$-S to $C_i$-S transition. Transport domains are shown in transparent light pink; scaffold domain in transparent green; helices H4c, H6b, and H9c in transparent purple; bound succinate in dark pink; lipid phosphorous atoms in gray; helices TM4b and H4c in cyan; helix HP$_{in}$a in dark green; helices TM9b and H9c in light orange; and helix HP$_{out}$a in dark orange. Structural alignment was performed using the scaffold domain's center of mass along the membrane-normal axis and using the transport domains' centers of mass along the membrane-parallel axes. https://elifesciences.org/articles/61350#video3

In addition to the noted substrate-release induced conformational changes, charge compensation is also essential to avoid slippage, ensuring Na$^+$-substrate coupling in DASS cotransporters. In fact, the Na$^+$ ions at the Na1 and Na2 sites in cotransporters can be regarded as equivalent to the cationic side chains of exchangers, though Na$^+$ reversibly binds. In cotransporters, the charge compensation model predicts that the transporter can only transition between $C_o$ and $C_i$ conformations when the transport domain is either fully-loaded or fully-unloaded. Such a mechanism ensures tightly coupled import of Na$^+$ and substrate in cotransporters, while the reversible binding of Na$^+$ allows for the concentration of the divalent substrate against its electrochemical gradient. Similar charge compensation mechanisms have been proposed for the citrate transporter CitS and the glutamate transporter EAAC1 (*Lolkema and Slotboom, 2017*; *Grewer et al., 2012*).

## Materials and methods

### Key resources table

| Reagent type (species) or resource | Designation | Source or reference | Identifiers | Additional information |
|---|---|---|---|---|
| Gene (*Lactococcus acidophilus*) | LaINDY | ENA | AAV42769.1 | |
| Gene (*Vibrio cholorea*) | VcINDY | ENA | AAF95939.1 | |
| Strain, strain background (*Escherichia coli*) | BL21(DE3) | Sigma-Aldrich | CMC0014 | |
| Strain, strain background (*Escherichia coli*) | JW2571 | Keio collection | JW2571 | |

*Continued on next page*

*Continued*

| Reagent type (species) or resource | Designation | Source or reference | Identifiers | Additional information |
|---|---|---|---|---|
| Strain, strain background (*Escherichia coli*) | 55244 | ATCC | 27C7 | |
| Recombinant DNA reagent | pET-LaINDY (plasmid) | This study | | See Materials and methods. To obtain the plasmid, contact the D.N. Wang Lab. |
| Recombinant DNA reagent | pET-VcINDY (plasmid) | *Mancusso et al., 2012* | | |
| Recombinant DNA reagent | pFab101 (plasmid) | *Miller et al., 2012* | | |
| Antibody (synthetic monoclonal) | Fab84 | This study | | See Materials and methods (3:1 molar ratio Fab:VcINDY). To obtain the Fab plasmid, contact the S. Koide Lab or the D.N. Wang Lab. |
| Chemical compound, drug | Amphipol | Anatrace | PMAL-C8 | |
| Software, algorithm | cryoSPARC | Structura Biotechnology | RRID:SCR_016501 | |
| Software, algorithm | Chimera | *Pettersen et al., 2004* | RRID:SCR_004097 | |
| Software, algorithm | PyMOL | Schrodinger | RRID:SCR_000305 | |
| Software, algorithm | COOT | *Emsley and Cowtan, 2004* | RRID:SCR_014222 | |
| Software, algorithm | PHENIX | *Adams et al., 2010* | RRID:SCR_014224 | |
| Software, algorithm | Prism | GraphPad Software | RRID:SCR_002798 | |
| Others | QuantiAu Foil R1.2/1.3 | Quantifoil | | |

## LaINDY transport activity assays in whole cells

As a target seed VcINDY was nominated to the cloning core of the New York Consortium of Membrane Protein Structure for the cloning of its homologs (*Love et al., 2010*). The homologous protein from *Lactobacillus acidophilus* (Uniprot: Q5FKK5_LACAC, LaINDY) was found to give the highest expression levels in *E. coli* BL21 DE3 cells when transformed with pET-LaINDY. The transport activity of LaINDY was characterized in *E. coli* whole cells following a published protocol with minor modifications (*Pos et al., 1998*; *Kim and Unden, 2007*). *E. coli* BL21 DE3 cells were transformed with a modified pET vector (*Love et al., 2010*) encoding N-terminal 10x His-tagged LaINDY (pET-LaINDY) and grown at 37°C until cells reached $OD_{595}$ of 0.7. Cells were induced with IPTG and growth was continued for 3 hr. Cells were harvested by centrifugation and resuspended at $OD_{595}$ = 10 in 10 mM NaCl, 100 mM choline chloride, 50 mM Tris pH 7.5. Cells were kept on ice until needed, and warmed to 30°C five mins prior to transport assay. The uptake reaction was initiated by addition of reaction buffer 10 mM NaCl, 100 mM choline chloride, 50 mM Tris pH 7.5, 1 μM $^3$H-succinate to the cell suspension at 1:10 (reaction buffer:cells) volumetric ratio. For the exchange reaction, 10 mM succinate or 10 mM α-ketoglutarate (αKG) was added 90 s after initiation of the uptake reaction. Aliquots were collected at fixed time points, with the reaction terminated by collecting the cells on pre-wetted 0.45 mm nitrocellulose filters mounted on a Hoeffer vacuum manifold and immediately washing with 4 mL ice cold 100 mM potassium phosphate buffer pH 7.5. The filters were incubated for 10 mins in scintillation fluid before measuring radioactivity using a Wallac 1450 Microbeta Plus liquid scintillation counter (Shelton, CT) (*Law et al., 2007*; *Law et al., 2008*; *Law et al., 2009*).

## *E. coli* growth assays with LaINDY complementation

The *E. coli* strain JW2571 (*Baba et al., 2006*), in which the only endogenous αKG transporter under aerobic conditions KgtP (*Seol and Shatkin, 1991*) was knocked out, was transformed with a pGEM-5Zf(+) vector encoding LaINDY. Transformed cells were grown in LB broth with ampicillin to an

$OD_{595}$ ~0.8, then diluted to an $OD_{595}$ of 0.1. Dilute cells were inoculated into eM9 media supplemented with 50 mM αKG (*Rhie et al., 2014*), and cell growth monitored using a Tecan SPECTRA-Fluor Plus microplate reader (Männedorf, Switzerland) incubated at 37°C.

## LaINDY expression and purification

LaINDY was expressed by autoinduction (*Studier and Moffatt, 1986*) at 25°C overnight in the *E. coli* strain BL21 DE3 transformed with the pET-LaINDY plasmid. Cells were harvested and lysed in a buffer of 50 mM Tris pH 8.0, 400 mM NaCl, 10 mM Imidazole. Ligand, either 10 mM $Na^+$ αKG or 10 mM $Na^+$ DL-malate, was added to the lysis buffer and all subsequent purification steps. Membranes were resuspended in a buffer of 50 mM Tris pH 8.0, 200 mM NaCl, 10 mM Imidazole, and solubilized in 1.2% dodecyl-maltoside (DDM), and protein was purified on a $Ni^{2+}$-NTA affinity column. The decahistidine tag was removed by overnight digestion at 25°C with TEV protease, followed by preparative size exclusion chromatography (SEC) in a buffer containing 25 mM Tris pH 8.0, 150 mM NaCl, 20% glycerol, and 0.075% DDM.

## Thermostabilization assay

A theremostability assay was used to search for compounds that stabilized LaINDY (*Auer et al., 2001*; *Mancusso et al., 2011*). Size exclusion chromatography purified LaINDY was dialyzed into a buffer of 20 mM Tris pH 8.0, 150 mM NaCl, 10% glycerol, 0.15% DM. Aliquots of 100 mg LaINDY were incubated with 100 mM test compounds at 42°C for 2 hr, and subsequently injected onto a Shodex KW804 analytical SEC column (Thomson, Clear Brook, VA) on HPLC (Shimadzu, Columbia, MA) in a buffer containing 200 mM $Na_2SO_4$, 50 mM Tris 7.5, 3 mM $NaN_3$, and 0.05% DDM. The height of the SEC peak for the 4°C control was used as a reference for normalization.

## Multi-angle dynamic light scattering

Purified LaINDY sample (50 μL) was injected onto a Shodex KW803 analytical SEC column on a Waters HPLC (Milford, MA) and eluted with the buffer containing 0.05% DDM at a rate of 0.5 mL/min. The mass of the LaINDY protein was determined using a Wyatt miniDAWN TREOS three angle-static light scattering detector (Santa Barbara, CA), a Wyatt Optilab rEX refractive index detector and a Waters 2489 UV absorbance detector (*Slotboom et al., 2008*; *Waight et al., 2010*). The differential refractive index ($dn/dc$) for DDM, 0.128 mL/g, was calculated using the refractive index detector. The size of the protein–detergent conjugate was deconvoluted following the published method (*Kendrick et al., 2001*), in which contributions from co-purifying lipids were not distinguished from those of the detergent.

## VcINDY expression and purification

Expression and purification of VcINDY was carried out according to our previous protocol (*Mancusso et al., 2012*). Briefly, *E. coli* BL21-AI cells (Invitrogen) were transformed with a modified pET vector (*Love et al., 2010*) encoding N-terminal 10x His tagged VcINDY. Cells were grown at 30°C until $OD_{595}$ reached 0.8, protein expression occurred at 19°C following IPTG induction, and cells were harvested 16 hr post-induction. Cell membranes were solubilized in 1.2% DDM and the protein was purified on a $Ni^{2+}$-NTA column. VcINDY was further purified by SEC in buffer containing 25 mM Tris pH 8, 100 mM NaCl, 10 mM $Na^+$-succinate, 5% glycerol and 0.075% DDM unless otherwise indicated.

## Nanodisc reconstitution

SEC purified His-tagged VcINDY protein was reconstituted into MSP2N2 nanodiscs (*Grinkova et al., 2010*) at a molar ratio of VcINDY: MSP: lipids of 1 (9 mM): 8 (72 mM): 277 (2.5 mM) (*Bayburt and Sligar, 2010*; *Schuler et al., 2013*). *E. coli* polar lipids in chloroform (Avanti) were vacuum-dried and rehydrated in nanodisc buffer containing 20 mM Tris pH 7.4, 100 mM NaCl, 0.5 mM EDTA and 0.1 mM TCEP to a concentration of 20 mg/mL. VcINDY, lipids and MSP2N2 protein were mixed in the nanodisc buffer to a final volume of 500 mL. Subsequently, 600 mg of Bio-Beads were added and incubated at 4°C overnight. The Bio-Beads were then removed and the solution was filtered through a 0.2 mm centrifugal filter. The nanodiscs containing VcINDY were purified by $Ni^{2+}$-NTA chromatography. The sample was incubated with a 1:4 molar ratio of Fab84 for 1 hr at 4°C before SEC.

## Fab development

The plasmid pFab101, a modified pFab007 (*Miller et al., 2012*) which includes a N31T mutation in the VH region, was used as the template to construct a Fab phage-display library termed NYC1 containing $1.1 \times 10^{11}$ sequences, following a previously described design (*Miller et al., 2012*). Fab library sorting was performed as previously described (*Fellouse et al., 2007*; *Dominik and Kossiakoff, 2015*) with minor modifications. In each round, phage solution was prepared in 50 mM Tris HCl buffer pH 7.5 containing 100 mM NaCl, 1% bovine serum albumin and 0.1 mM TCEP with or without 2 mM succinate, and was first incubated with streptavidin-coated magnetic beads harboring biotinylated nanodisc that did not contain an embedded protein ('empty' nanodisc). The supernatant of this reaction was incubated with 100 nM VcINDY embedded in biotinylated nanodisc, and phages bound to the VcINDY-nanodisc complex were captured using streptavidin-coated magnetic beads. A total of four rounds of library sorting were performed. Enriched clones were individually tested using phage ELISA (*Fellouse et al., 2007*; *Sidhu et al., 2000*).

## Fab expression and purification

Target Fab84 was subcloned into the Fab expression vector Ptac_Fab_accept_tagless (*Burioni et al., 1998*). After rigidification (*Bailey et al., 2018*), the subsequent Fab-containing plasmid was transformed into *E. coli* 55244 cells. Cells were grown in TGB media at 30°C for 22 hr and harvested. Fab84 protein was purified by injection onto a Protein G column (GE healthcare) and eluted with 100 mM glycine-pH 2.7. Protein was collected in 1 mL fractions containing 2 M Tris pH 8. Fractions containing protein were dialyzed against 50 mM sodium acetate pH 5.3 before purification using a Resource S column (GE healthcare).

## Amphipol exchange

After nickel column purification PMAL-C8 (Anatrace, Maumee, OH) was added to the detergent-purified transporter protein at a 1:5 protein:amphipol weight ratio (*Huynh et al., 2018*). The mixture was incubated at 4° C overnight with nutating. To remove detergent, Bio-Beads were incubated with sample at a 1:100 detergent:Bio-Beads weight ratio for 2 hr at 4°C with gentle agitation. The Bio-Beads were then removed by centrifugation at 4,500 rpm and the sample was further purified by SEC in buffer containing 25 mM Tris pH 7.5, 100 mM NaCl and 0.1 mM TCEP.

## Cryo-EM sample preparation and data collection

All cryo-EM grids were prepared by applying 3 mL of protein at ~3 mg/mL to a glow-discharged QuantiAuFoil R1.2/1.3 300-mesh grid (Quantifoil) and blotted for 2.5 to 4 s under 100% humidity at 4°C before plunging into liquid ethane using a Mark IV Vitrobot (FEI).

Cryo-EM data of apo LaINDY were acquired on a Titan Krios microscope with a K2 direct electron detector (Gatan), using a GIF-Quantum energy filter with a 15 eV slit width. Leginon (*Suloway et al., 2005*) was used for ice thickness targeting and automated data collection. Each micrograph was dose-fractioned over 50 frames, with an accumulated dose of 75 $e^-/Å^2$.

Cryo-EM data of LaINDY-malate were acquired on a Titan Krios microscope with a K2 direct electron detector, using a GIF-Quantum energy filter with a 20 eV slit width. SerialEM was used for automated data collection (*Schorb et al., 2019*). Each micrograph was dose-fractioned over 60 frames, with an accumulated dose of 50 $e^-/Å^2$.

Cryo-EM data of LaINDY-αKG were acquired on a Talos Arctica microscope with a K3 direct electron detector. Leginon was used for ice thickness targeting and automated data collection (*Rice et al., 2018*). Each micrograph was dose-fractioned over 56 frames, with an accumulated dose of 46 $e^-/Å^2$.

Cryo-EM data of VcINDY-Na$^+$-Fab84 in nanodiscs were acquired on a Talos Arctica microscope with a K2 direct electron detector. SerialEM was used for automated data collection. Each micrograph was dose-fractioned over 50 frames, with an accumulated dose of 44 $e^-/Å^2$.

Cryo-EM data of VcINDY-Na$^+$ in amphipol were acquired on a Talos Arctica microscope with a K2 direct electron detector. SerialEM was used for automated data collection. Each micrograph was dose-fractioned over 48 frames, with an accumulated dose of 40 $e^-/Å^2$.

## Cryo-EM image processing

Motion correction, CTF estimation, particle picking, 2D classification, *ab initio* model generation, heterogenous and non-uniform refinement, and per particle CTF refinement were all performed with cryoSPARC (*Punjani et al., 2017*). Each dataset was processed using the same protocol, except as noted. All maps were sharpened using Auto-sharpen Map in Phenix (*Adams et al., 2010*).

Micrographs underwent motion correction and the CTF estimated, and those with an overall resolution worse than 8 Å excluded from subsequent steps. An ellipse-based particle picker identified particles used to generate initial 2D classes. These classes were used for template-based particle picking. Template identified particles were extracted and subjected to 2D classification. A subset of well resolved 2D classes were used for the initial *ab initio* model building, while all picked particles were subsequent used for heterogeneous 3D refinement. After multiple rounds of 3D classification (*ab initio* model generation and heterogeneous 3D refinement with two or more classes), a single class was selected for non-uniform 3D refinement, resulting in the final map.

For the VcINDY-Na$^+$-Fab84 sample in nanodiscs, refinement of an initial 3.36 Å map indicated oblique views were rare in the particle set. Projections of this map were therefore used as templates for an additional round of particle picking, with subsequent 3D classification, per particle CTF refinement and non-uniform 3D refinement.

## LaINDY crystallization and X-ray diffraction data collection

LaINDY protein purified in NG and αKG was concentrated to 2 mg/mL using a centrifugal filtration device and crystallized at 18°C using the hanging drop vapor diffusion method by mixing equal volumes of concentrated protein and well solution of 30% Jeffamine ED-2001, 100 mM HEPES pH 6.8. Crystal quality was improved by the addition of 10 mM malic acid prior to size exclusion chromatography. Crystals were flash frozen in liquid nitrogen with crystallization solution serving as cryoprotectant. Crystals were of the P2$_1$ space group with unit cell dimensions of a = 91.3 Å, b = 76.6 Å, c = 96.9 Å, β = 90.5° and contained two molecules in the asymmetric unit, which form a single transporter dimer. Data were collected at the Advanced Light Source beamline 5.0.2. Diffraction data were processed and scaled in HKL2000 (*Otwinowski and Minor, 1997*).

## VcINDY crystallization and X-ray diffraction data collection

VcINDY purified in DM was incubated with 10 mM sodium terephthalate (TTP) on ice for 30 min prior to crystallization with a protein concentration of 4 mg/mL at 4°C by hanging-drop vapor diffusion. Crystals were grown in 30% PEG 550, 50 mM NaCl and 100 mM Tris pH 8.0, and were subsequently frozen in liquid nitrogen with the crystallization solution serving as the cryoprotectant. X-ray data were collected at the Advanced Light Source Beamline 5.0.2. Data processing and scaling were performed using HKL2000. Crystals were of space group P2$_1$ with unit cell dimensions around a = 108 Å, b = 103 Å, c = 174 Å, β = 96°, and contained four molecules per asymmetric unit.

## Model building and refinement

Cryo-EM models were built in Coot (*Emsley and Cowtan, 2004*) and refined in Phenix real space refine (*Adams et al., 2010*). The LaINDY-apo model was manually built, and subsequently used as an initial model for the LaINDY-malate and LaINDY-αKG structures. VcINDY models were built using the VcINDY X-ray structure (PDB: 5UL9), with sodium and citrate removed, as an initial model. The Fab was rebuilt from an initial model of a homologous X-ray structure (PDB: 5EII) with the variable loops removed. In both VcINDY and LaINDY maps some discontinuous densities could be attributed to the aliphatic chains of lipids, detergents, or amphipols and were modeled as alkanes for simplicity.

The LaINDY X-ray structure was determined by molecular replacement using the LaINDY-apo cryo-EM model as a starting model, followed by model building in Coot and refinement in Phenix.

The VcINDY-TTP X-ray structure was determined by molecular replacement using the structure of VcINDY (PDB ID: 5UL9), with sodium and citrate removed, as the initial search model. The model was refined using group B-factors, and NCS, secondary structure and reference model restraints.

### Phylogenetic tree and sequence conservation

DASS family orthologs were aligned in Promals3D (*Pei et al., 2008*), including VcINDY (PDB ID: 5UL9) and CitS (PDB ID: 5A1S) as structural models. As a non-member of the DASS family, CitS was used as the outgroup when calculating distances using FastTree (*Price et al., 2009*).

### Bond length analysis

The distance between $C_\alpha$ and C carbons was calculated using a custom Biopython script (*Cock et al., 2009*). Median $C_\alpha$ to C distances were calculated for 2045 single particle cryo-EM and 64,676 X-ray structures deposited in the Protein Data Bank between 2013 and 2020, containing only L-type polypeptides, and resolved to at least 4 Å.

### System preparation

Molecular dynamics (MD) simulation systems of the $C_o$-S and $C_o$ states of LaINDY were constructed from its X-ray crystal structure. Succinate was added to the X-ray crystal structure of LaINDY by aligning the cryo-EM structure of LaINDY with $\alpha$-ketoglutarate bound and mutating the $\alpha$-ketoglutarate into succinate. Force field parameters for succinate were determined using the online CGᴇɴFF interface (*Vanommeslaeghe and MacKerell, 2012*; *Vanommeslaeghe et al., 2012*). The protonation states of the titratable residues of LaINDY were determined using PROPKA (*Søndergaard et al., 2011*; *Olsson et al., 2011*). Two independent POPE (1-palmitoyl-2-oleoyl-sn-glycero-3-phosphoethanolamine) membranes were prepared using the Mᴇᴍʙʀᴀɴᴇ Bᴜɪʟᴅᴇʀ (*Wu et al., 2014*; *Jo et al., 2009*; *Jo et al., 2007*; *Lee et al., 2019*) tool of CHARMM-GUI (*Jo et al., 2008*). The orientation of LaINDY in the membrane was determined by aligning the protein's symmetry axis with the membrane-normal axis, and the insertion depth in the membrane was determined by applying the PPM (Pᴏsɪᴛɪᴏɴɪɴɢ ᴏꜰ Pʀᴏᴛᴇɪɴs ɪɴ Mᴇᴍʙʀᴀɴᴇ) Sᴇʀᴠᴇʀ (*Lomize et al., 2012*) to a model (*Mulligan et al., 2016*) of the $C_o$-Na$^+$-S state of VcINDY. Once the membrane and protein were combined, all lipids within 1 Å of the protein were removed from the systems. Water was added to the systems using the VMD (*Humphrey et al., 1996*) Sᴏʟᴠᴀᴛᴇ plugin. The systems were neutralized with Cl$^-$, and 0.15 M NaCl was added to the systems using the VMD Aᴜᴛᴏɪᴏɴɪᴢᴇ plugin.

### Simulation parameters

All simulations were performed with NAMD (*Phillips et al., 2005*) using the CHARMM36m (*Huang et al., 2017*; *Klauda et al., 2010*; *Vanommeslaeghe et al., 2010*; *Yu et al., 2012*; *Jorgensen et al., 1983*; *Venable et al., 2013*) force field at a constant temperature of 310 K and a constant pressure of 1 atm. Constant temperature was maintained using Langevin dynamics for all non-hydrogen atoms with a damping coefficient of 1 ps$^{-1}$. Constant pressure was maintained using a Nosé-Hoover Langevin piston (*Martyna et al., 1994*; *Feller et al., 1995*) with a period of 100 fs and a damping time scale of 50 fs. The cut-off distance for both electrostatic and van der Waals interactions was set to 12 Å, and a switching function was applied at 10 Å. Periodic boundary conditions were applied in all simulations, and long-range electrostatic interactions were calculated using the particle mesh Ewald method (*Darden et al., 1993*; *Essmann et al., 1995*) with a grid point density of 1 Å$^{-1}$.

### Simulations performed

After building the systems, harmonic restraints were applied to the constituent atoms of the protein, membrane, and bound succinate (i.e., only the bulk water and ions were left unrestrained). A force constant of 1.0 kcal/mol/Å$^2$ was used, and an integration time step of 1 fs was used. Three thousand steps of conjugate gradient energy minimization were performed, after which the systems were simulated with MD for 1 ns. The restraints on the membrane were then released, and the systems were minimized for three thousand steps and then simulated for 10 ns. Next, the restraints on the protein and bound succinate were released, and the systems were minimized for three thousand steps and then simulated for 10 ns. With initial equilibration complete, the integration time step was changed to 2 fs, and an additional 100 ns of unbiased equilibrium simulation was performed for each system to sample the dynamics of the $C_o$-S and $C_o$ states of LaINDY.

Biases were then applied with the goal of inducing a transition from the $C_o$-S state to an approximate target model of the $C_i$-S state. These biases and the approximate $C_i$-S target are described in

the 'Biasing Protocol' section below. To simplify comparison between the transitions sampled by the two unique protomers of LaINDY and between independent trials of the simulation, a 10 ns transition was first induced to a convenient reference $C_o$-S state: the X-ray crystal structure. The system was then restrained at this reference $C_o$-S state for 10 ns to allow the membrane and aqueous environment to equilibrate around this new protein state. Next, a 100 ns transition to the approximate $C_i$-S target was induced, and the system was subsequently restrained at the approximate $C_i$-S target for 50 ns again to allow the environment to equilibrate around the new protein state. Finally, all restraints were released, and a 100 ns unbiased equilibrium simulation was performed to examine the stability of and to sample the dynamics of the newly generated model of the $C_i$-S state of LaINDY. See *Figure 6—figure supplement 1a* for a graphical overview of the simulations performed.

The transition to the $C_i$-S state was first induced without any biases applied to the bound succinate. A successful transition was sampled by protomer B, but succinate spontaneously unbound during the early stages of the transition sampled by protomer A. To increase sampling, an additional trial of this transition was performed with biases applied to the bound succinate (see 'Biasing Protocol'). Throughout this article, data from the simulation without biases applied to succinate are labeled 'protomer $B_1$.' Data from the simulation with biases applied to succinate are labeled 'protomer $A_2$' and 'protomer $B_2$.'.

## Biasing protocol

In transporters with an inverted repeat topology, it has been shown that the $C_o$-S and $C_i$-S states can be generated from each other by swapping the conformations of the transporter's repeats (*Forrest et al., 2011*). The internal symmetry in these transporters means that the fundamental global structural differences between their $C_o$-S and $C_i$-S states can be observed simply by rotating one state 180° about any axis in the plane of the membrane. When this approach is applied to LaINDY, it is immediately apparent that the fundamental global differences between the $C_o$-S X-ray crystal structure and the newly generated approximate $C_i$-S target model are: (1) the orientation of the transport domains relative to the scaffold domain and (2) the position of the transport domains relative to the scaffold domain along the membrane-normal axis. To describe these differences quantitatively and to induce a transition to the approximate $C_i$-S target, the Collective Variables (Colvars) module (*Fiorin et al., 2013*) in NAMD was used.

In the Colvars module, collective variables are mathematical functions used to describe collective properties of interest of selections of atoms within a system, and transitions are induced by applying moving harmonic potentials to the collective variables. In this case, one collective variable was defined for each of the fundamental global differences observed between the $C_o$-S state and approximate $C_i$-S target. Specifically, for the orientations of the transport domains, spinAngle collective variables were used, and distanceZ collective variables were used for the positions of the transport domains relative to the scaffold domain along the membrane-normal axis. Helix $C_\alpha$ atoms of the transport domains and transmembrane helix $C_\alpha$ atoms of the scaffold domain were used in the definitions of these collective variables. Schematic depictions of these collective variables are shown in *Figure 6—figure supplement 1b*.

To determine the rotation axis used in the definition of the spinAngle collective variables, the rotation matrix needed to structurally align the transport domains of the $C_o$-S state with the transport domains of the approximate $C_i$-S target was calculated using the measure fit command of VMD. Importantly, the order option was used to allow the first repeat of the $C_o$-S state to be compared to the second repeat of the approximate $C_i$-S target at the same time that the second repeat of the $C_o$-S state was compared to the first repeat of the approximate $C_i$-S target. This calculation also determined the change in angle (i.e., amount of spin about the spin axis) that was needed to induce the transition to the approximate $C_i$-S target. To induce the transition to the approximate $C_i$-S target using biased MD simulations, moving harmonic restraints with a force constant of 8 kcal/mol/(°)$^2$ were applied to the spinAngle collective variables.

Calculating the change in the distanceZ collective variables required to induce the transition to the approximate $C_i$-S target was straightforward, as the approach used to generate the approximate $C_i$-S target simply changes the direction of the displacement along the membrane-normal axis of the transport domains relative to the scaffold domain without affecting its magnitude. Moving harmonic restraints with a force constant of 80 kcal/mol/Å$^2$ were applied to the distanceZ collective variables.

*Video 1* demonstrates how the approximate $C_i$-S target was generated, and it shows the VMD-based quantification (described above) of the global structural differences between the $C_o$-S state and the approximate $C_i$-S target used to define the spinAngle and distanceZ collective variables.

In addition to the spinAngle and distanceZ collective variables, harmonic restraints were applied to up to two additional collective variables during the induced transition. First, the orientation of the scaffold domain was restrained to prevent the overall orientation of LaINDY from changing. Specifically, orientation collective variables were used on the scaffold domains (defined as above), and harmonic restraints with a force constant of 100,000 kcal/mol were applied to them. Second, in one trial of the transition (see 'Simulations Performed'), succinate was restrained to the binding sites of the transport domains using distance collective variables. One distance collective variable per succinate was defined using the non-hydrogen atoms of the succinate and the $C_\alpha$ atoms of the residues that were consistently closest to the bound succinate during the unbiased simulations of the $C_o$-S state of LaINDY (i.e., Pro154-Arg159, Thr208-Pro212, Ala396-Thr399, and Asn442-Pro444). A half-harmonic restraint centered at 2.0 Å and with a force constant of 1.0 kcal/mol/Å$^2$ was applied to these collective variables.

## Data and code availability

Cryo-EM maps and models have been deposited in the Protein Data Bank and EMDB database, respectively, for VcINDY-Na$^+$ in amphipol (6WU3, EMD-21904), VcINDY-Na$^+$-Fab84 in nanodisc (6WW5, EMD-21928), LaINDY-apo (6WU1, EMD-21902), LaINDY-αKG (6WU4, EMD-21905), and LaINDY-malate (6WU2, EMD-21903). X-ray derived models and diffraction data have been deposited in the Protein Data Bank for LaINDY-malate-αKG (6WTW) and VcINDY-TTP (6WTX). Coordinates of representative LaINDY structures from MD simulations of the $C_o$-S and $C_i$-S states have been made publicly available on Zenodo (DOI 10.5281/zenodo.3965996). Bond length analysis code is available at *Sauer, 2020*; https://github.com/DavidBSauer/bond_length_analysis (copy archived at https://github.com/elifesciences-publications/bond_length_analysis).

## Acknowledgements

This work was financially supported by the NIH (R01NS108151, R01GM121994 and R01DK099023 to DNW; U54GM095315 to WA Hendrickson; P41GM104601, R01GM067887 and R01GM123455 to ET), the TESS Research Foundation and the American Epilepsy Society (to DNW). DBS was supported by the American Cancer Society Postdoctoral Fellowship (129844-PF-17-135-01-TBE) and Department of Defense Horizon Award (W81XWH-16-1-0153). NT was supported by a NSF Graduate Research Fellowship (1746047). NC was supported by an NIH Predoctoral Training Grant (T32-GM088118). VcINDY homologs were cloned by B Kloss at the New York Consortium of Membrane Protein Structures. We thank the following colleagues for reagent, technical assistance, and helpful discussions: JP Armache, JG Belasco, N Coudray, T Hattori, J Jiang, NK Karpowich, M Lopez Redondo, Z Liu, R Mancusso, AB Rejto and SG Sligar. We are also grateful to the staff at the following facilities for assistance in screening and data collection in cryo-EM and X-ray diffraction: the NYU Cryo-EM Facility, the Pacific Northwest Center for Cryo-EM, Advanced Light Source Beamline 5.0.2 at the Berkeley National Laboratory, beamlines AMX and FMX at NSLS-II and 19-BM and 19-ID at the Advanced Photon Source. EM data processing used computing resources at the HPC Facility of NYULMC, and we were assisted by A Siavosh-Haghighi and M Costantino. MD simulations were performed using supercomputing allocations from Blue Waters and XSEDE (TG-MCA06N060) to ET. Blue Waters is supported by the NSF (OCI-0725070 and ACI-1238993), the State of Illinois and the National Geospatial-Intelligence Agency. XSEDE is supported by the NSF (ACI-1548562).

## Additional information

### Funding

| Funder | Grant reference number | Author |
| --- | --- | --- |
| National Institutes of Health | R01NS108151 | Da-Neng Wang |
| National Institutes of Health | R01GM121994 | Da-Neng Wang |

| National Institutes of Health | R01DK099023 | Da-Neng Wang |
| National Institutes of Health | U54GM095315 | Da-Neng Wang |
| National Institutes of Health | P41GM104601 | Emad Tajkhorshid |
| National Institutes of Health | R01GM067887 | Emad Tajkhorshid |
| TESS Research Foundation | | Da-Neng Wang |
| American Epilepsy Society | AES2017SD3 | Da-Neng Wang |
| American Cancer Society | 129844-PF-17-135-01-TBE | David B Sauer |
| Department of Defense | W81XWH-16-1-0153 | David B Sauer |
| National Science Foundation | 1746047 | Noah Trebesch |
| National Institutes of Health | T32GM088118 | Nicolette Cocco |
| XSEDE | TG-MCA06N060 | Emad Tajkhorshid |
| National Institutes of Health | R01GM123455 | Emad Tajkhorshid |
| Blue Waters | | Emad Tajkhorshid |

The funders had no role in study design, data collection and interpretation, or the decision to submit the work for publication.

## Author contributions

David B Sauer, Noah Trebesch, Data curation, Software, Formal analysis, Funding acquisition, Validation, Investigation, Visualization, Methodology, Writing - original draft, Writing - review and editing; Jennifer J Marden, Nicolette Cocco, Jinmei Song, Akiko Koide, Data curation, Formal analysis, Investigation, Writing - review and editing; Shohei Koide, Resources, Data curation, Formal analysis, Supervision, Funding acquisition, Investigation, Project administration, Writing - review and editing; Emad Tajkhorshid, Conceptualization, Resources, Data curation, Software, Formal analysis, Supervision, Funding acquisition, Validation, Investigation, Methodology, Writing - original draft, Project administration, Writing - review and editing; Da-Neng Wang, Conceptualization, Formal analysis, Supervision, Funding acquisition, Methodology, Writing - original draft, Project administration, Writing - review and editing

## Author ORCIDs

David B Sauer http://orcid.org/0000-0001-9291-4640
Noah Trebesch http://orcid.org/0000-0001-5536-4862
Emad Tajkhorshid https://orcid.org/0000-0001-8434-1010
Da-Neng Wang https://orcid.org/0000-0002-6496-4699

## Decision letter and Author response

Decision letter https://doi.org/10.7554/eLife.61350.sa1
Author response https://doi.org/10.7554/eLife.61350.sa2

# Additional files

## Supplementary files

- Transparent reporting form

## Data availability

Cryo-EM maps and models have been deposited in the Protein Data Bank and EMDB database, respectively, for VcINDY-Na+ in amphipol (6WU3, EMD-21904), VcINDY-Na+-Fab84 in nanodisc (6WW5, EMD-21928), LaINDY-apo (6WU1, EMD-21902), LaINDY-aKG (6WU4, EMD-21905), and LaINDY-malate (6WU2, EMD-21903). X-ray derived models and diffraction data have been deposited in the Protein Data Bank for LaINDY-malate-aKG (6WTW) and VcINDY-TTP (6WTX). Coordinates of representative LaINDY structures from MD simulations of the Co-S and Ci-S states have been made

publicly available on Zenodo (DOI: 10.5281/zenodo.3965996). Bond length analysis code is available at https://github.com/DavidBSauer/bond_length_analysis (copy archived at https://github.com/elifesciences-publications/bond_length_analysis).

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

## Appendix 1

## Supplementary Note

### Molecular Dynamics Simulations

Using MD, we simulated the unbiased equilibrium dynamics of the $C_o$-S state of LaINDY, we induced a transition to an approximate target model of the $C_i$-S state using a custom biased MD simulation protocol, and we simulated the equilibrium dynamics of the new putative $C_i$-S state (*Video 2*). In the custom biased MD simulation protocol, biases were applied to collective variables representing the displacement $z$ and orientation $\theta$ of the transport domains of LaINDY relative to its scaffold domain (*Figure 6—figure supplement 1b*). From the ensembles generated by the unbiased equilibrium MD simulations of the $C_o$-S and putative $C_i$-S states, we selected as representative the structures with the most representative values of $z$ and $\theta$ concurrently in both protomers (*Figure 6—figure supplement 1i*). Water accessibility profiles of these structures demonstrate that the new state is definitively a $C_i$-S state (*Figure 6—figure supplement 1j*), and $z$ and $\theta$ time series indicate that the new $C_i$-S state is stable (*Figure 6—figure supplement 1c&d*).

To generate a model of a transporter's missing state based on its internal inverted repeat topological symmetry, standard procedure is to apply homology modeling using the two repeats as structural templates for each other (*Vergara-Jaque et al., 2015*). To generate our model of the $C_i$-S state of LaINDY, we instead used a novel MD-based approach that only indirectly drives a swap of the conformations of LaINDY's inverted repeats (*Figure 6—figure supplement 1e,f &k*), and we favor this approach for several reasons. Most importantly, the model is generated in the presence of its natural membrane environment, and confirmation of its stability in this environment via MD is built in to the approach. This approach also enables the transporter dynamics to be assessed before, after, and during the transition between states, and it lays the groundwork needed to apply an advanced array of MD-based techniques to characterize detailed structural and thermodynamic properties of the transition (*Moradi and Tajkhorshid, 2013*; *Moradi and Tajkhorshid, 2014*; *Moradi et al., 2015*).

Because the structure of VcINDY's $C_i$-$Na^+$-S state has been determined, it also would have been possible to generate a model of the $C_i$-S state of LaINDY using homology modeling with VcINDY as a template. We favor our MD-based approach over this second homology modeling-based approach for the same reasons listed in the previous paragraph. That said, it would have been possible to use information from the $C_i$-$Na^+$-S state of VcINDY, rather than information from LaINDY's inverted repeats, to generate $C_i$-S targets for the biased MD simulations. Within the collective variable space, the targets suggested by LaINDY's inverted repeats are $-4.16$ Å for $z$ and $-19.5°$ for $\theta$, and the targets suggested by VcINDY's $C_i$-$Na^+$-S state are $-8.34$ Å for $z$ and $-22.1°$ for $\theta$. Both sets of targets are reasonable, but we ultimately chose to use the targets provided by the inverted repeats because LaINDY and VcINDY, while homologous, are different proteins. The stability of the LaINDY $C_i$-S model at the $z$ and $\theta$ targets (*Figure 6—figure supplement 1c&d*) and the accessibility of bound succinate from the cytosol (*Figure 6—figure supplement 1j*) demonstrate the quality of the inverted repeat-based model, but we cannot comment on what the results would have been if the VcINDY-based targets had been used instead.

The biased MD simulations we have performed are a powerful tool for assessing LaINDY's dynamics during the transition between its $C_o$-S and $C_i$-S states, but they also have limitations. Importantly, the pathway taken through the collective variable space during the transition has not been optimized in these simulations. As such, we restricted analyses of our biased MD simulations to the global structural differences between the end states and the internal dynamics of the transport domains, which this work suggests move as rigid bodies relative to the scaffold domain. Because restraints were applied to succinate in simulation trials $A_2$ and $B_2$, the internal structural dynamics of the transport domains could have been perturbed in these simulations, but we suggest that this did not happen because of the consistency of the data from these trials with the data from simulation trial $B_1$, in which no restraints were applied to succinate. Additionally, the biased MD simulations we have performed cannot be used to generate free energy profiles associated with the transitions, also limiting the kinds of analyses we were able to perform. To overcome these limitations and expand on this work, an advanced array of MD-based techniques would need to be applied to the transition

(*Moradi and Tajkhorshid, 2013*; *Moradi and Tajkhorshid, 2014*; *Moradi et al., 2015*). We again note that the biased MD simulations we have performed lay the groundwork needed to apply these advanced MD-based techniques.

