## [Decision Letter]

**Acceptance summary:**

This manuscript makes an important contribution to the membrane-transporter field by presenting high-resolution structures of four different conformational states of the divalent anion sodium symporter (DASS) family. They authors complement their structural studies with molecular dynamics simulations. Overall, the results present a rational framework for understanding the mechanism of the transport cycle.

**Decision letter after peer review:**

Thank you for submitting your article "Structural basis for the reaction cycle of DASS dicarboxylate transporters" for consideration by *eLife*. Your article has been reviewed by three peer reviewers, and the evaluation has been overseen by Merritt Maduke as the Reviewing Editor and Richard Aldrich as the Senior Editor. The following individuals involved in review of your submission have agreed to reveal their identity: Raimund Dutzler (Reviewer #1); Hai Lin (Reviewer #2).

The reviewers have discussed the reviews with one another and the Reviewing Editor has drafted this decision to help you prepare a revised submission.

Summary:

In this study, the authors describe multiple hitherto unseen structures of prokaryotic transporters from the divalent anion sodium symporter (DASS) family, including a new structure of VcINDY, which is the model DASS co-transporter, in its Na^+^-only-bound state, and multiple structures of LaINDY, which is an antiporter. The VcIndy structures (in this manuscript and those published previously) adopt inward-facing conformations. To obtain information of unknown states on the transport cycle, the authors turned to the homologous protein LaINDY, which allowed structure determination of outward- facing conformations in presence and absence of substrate. Due to the homology of the proteins, the structures provide general models of the opposite endpoints during transport by the alternate access mechanism. To provide further insight into the trajectory of conformations connecting both states, the authors used molecular-dynamics simulations by combining the novel structure of the outward-facing state of LaINDY and a model of the inward-facing state that was constructed based on geometrical considerations and knowledge of the VcINDY structure. By the combination of experimental data and computer simulations, the authors propose a charge-compensation mechanism, together with specific protein conformational changes, to explain the substrate coupling. Overall, this well-written manuscript reveals new insights into the transport cycle of DASS transporters and will be of interest to other researchers.

Essential revisions:

1) The manuscript describes the transport mechanism of LaINDY as a coupled exchanger based on cellular uptake studies. On basis do you exclude a potential mechanism as uncoupled dicarboxylate transporter or even proton-coupled symporter? In the subsection “Structure determination of multiple states”, the authors present convincing data that LaINDY is capable of transporting succinate and αKG using both growth assays and cell-based transport assays. However, these transport data do not explicitly demonstrate that an exchange reaction is occurring as the authors state. The data closest to showing that exchange is occurring is Figure 2A where the addition of cold substrate induced efflux of accumulated radiolabelled succinate. However, this type of efflux of labelled substrate would also be observed for a symporter in this assay, which is basically a classical cold-chase assay. The phylogenetic data certainly suggest that LaINDY is an exchanger, but these transport assays are not conclusive. Perhaps the authors could address the need for more detailed mechanistic dissection of LaINDY to address this issue.

2) The authors have compared the bond-length distribution of Cα-C bonds between different structures and found the mean of the distribution to be somewhat shorter in their cryo-EM structures (except for the LaINDY-apo structure which was collected on a different microscope). Could this be due to the inaccuracy of the calibration of the microscope and the detector? In any case, it should be emphasized that although all datasets are all of high quality and clearly permit interpretation of sidechain density, the exact sidechain conformation cannot be unambiguously assigned at this resolution (which is a common problem of many membrane protein structures).

3) In the computer simulation of the transition between the outward-facing and inward-facing state of LaINDY, the authors have applied the following procedure: They started their simulations with the X-ray structure of the outward facing conformation of LaINDY and applied a bias potential towards a model of the inward-facing state of the same protein that was constructed by a protocol that exploits the inverted repeat architecture of the transporter subunit. Although this approach is generally elegant, would a homology model of the inward-facing state of LaINDY generated based on the inward-facing conformation of VcINDY as template, provide a more accurate description of such state? Please comment on the difference between the two models of inward-facing conformations.

4) Please discuss whether you have you spotted potential metastable intermediates or larger barriers of the transporter on the way between the two extreme states.

5) The use of biased MD is justified, as the time scale of the transport is way beyond what can be currently achieved. Nevertheless, the authors should remind the readers that the biases and restraints imposed in the simulations might have biased the results toward the proposed transport routes.

6) It would be of interest to see whether the conformation of LaINDY will change if the mutations are made to either/both of the Arg159/Glu146 and His392/His401 pairs in the equivalent Na-sites.

7) Figure 1—figure supplement 2. Clones #76, 78 and 83 are said to be identical, but have different A450 readings for Na^+^ and Na^+^ + succinate states in Figure 1—figure supplement 2. If they are identical, how this is possible, especially as #78 has a higher value for T5 than T6, whereas #76 and #83 have the opposite pattern?

8) Subsection “Structure determination of multiple states” and Figure 2. The authors convincingly showed that succinate can be transported by LaINDY, but they observe little/no succinate-induced stabilisation. Is 10 mM substrate not sufficient to stabilise the protein? Or do the authors anticipate a low binding affinity for succinate under these conditions?

9) Subsection “Structure determination of multiple states”. Please explain the use of terephthalate instead of succinate in this X-ray structure. Is there a rationale for this?

10) The authors identify the "LaINDY handle", but do not mention it again. Do the authors think it is mechanistically significant?

11) Subsection “LaINDY has Na^+^ surrogate side-chains near the substrate binding site”. The authors only mention charged residue surrogates for Na1 and Na2, but VcINDY requires 3 Na^+^ ions. Does LaINDY offer up any clues as to the location of the 3rd Na^+^ site? There is 10 mM Na^+^ in the reaction buffer in the transport assay. Is it certain that LaINDY (or other DASS exchangers) is Na^+^-independent?

12) Subsection “Substrate release from VcINDY causes significant structural changes”. The authors compare the previous detergent-solubilised VcINDY structures that have Na^+^ and succinate bound to the amphipol solubilised VcINDY with Na^+^ only bound and highlight prominent changes to periplasmic surface. Are the authors suggesting that these changes are due to the substrate bound state of the protein? If so, is it not possible that these changes are merely due to a detergent vs. amphipol effect?

---

## [Author Response]

Essential revisions:1) The manuscript describes the transport mechanism of LaINDY as a coupled exchanger based on cellular uptake studies. On basis do you exclude a potential mechanism as uncoupled dicarboxylate transporter or even proton-coupled symporter? In the subsection “Structure determination of multiple states”, the authors present convincing data that LaINDY is capable of transporting succinate and αKG using both growth assays and cell-based transport assays. However, these transport data do not explicitly demonstrate that an exchange reaction is occurring as the authors state. The data closest to showing that exchange is occurring is Figure 2A where the addition of cold substrate induced efflux of accumulated radiolabelled succinate. However, this type of efflux of labelled substrate would also be observed for a symporter in this assay, which is basically a classical cold-chase assay. The phylogenetic data certainly suggest that LaINDY is an exchanger, but these transport assays are not conclusive. Perhaps the authors could address the need for more detailed mechanistic dissection of LaINDY to address this issue.

We agree with the reviewers that our description of LaINDY as an exchanger is not certain. And the reviewers rightly point out that demonstration of exchange is non-trivial, as cotransporters can catalyze similar reactions. As the reviewers suggested, we have now clarified this point in the Discussion and emphasized that further experiments in reconstituted proteoliposomes are needed (Discussion, second paragraph).

2) The authors have compared the bond-length distribution of Cα-C bonds between different structures and found the mean of the distribution to be somewhat shorter in their cryo-EM structures (except for the LaINDY-apo structure which was collected on a different microscope). Could this be due to the inaccuracy of the calibration of the microscope and the detector? In any case, it should be emphasized that although all datasets are all of high quality and clearly permit interpretation of sidechain density, the exact sidechain conformation cannot be unambiguously assigned at this resolution (which is a common problem of many membrane protein structures).

The reviewers have pinpointed the exact reason we did the Cα-C bond length calculation, to ensure microscope and camera calibration did not affect our conclusions. However, we have come to think that the observed discrepancy in bond length between X-ray and cryo-EM is not due to inaccuracy in microscope calibration, as the cryo-EM bond lengths are *systematically* smaller than X-ray and ideal, both in our structures and PDB depositions. If microscope calibration were an issue, we would expect the direction of this error to be random, changing the variance of the distribution but not of the mean value of structures’ bond lengths.

We thank the reviewers for pointing out that this length difference is small and does not affect the scientific conclusions drawn from our structures. We have emphasized this in the subsection “Structure determination of multiple states”.

3) In the computer simulation of the transition between the outward-facing and inward-facing state of LaINDY, the authors have applied the following procedure: They started their simulations with the X-ray structure of the outward facing conformation of LaINDY and applied a bias potential towards a model of the inward-facing state of the same protein that was constructed by a protocol that exploits the inverted repeat architecture of the transporter subunit. Although this approach is generally elegant, would a homology model of the inward-facing state of LaINDY generated based on the inward-facing conformation of VcINDY as template, provide a more accurate description of such state? Please comment on the difference between the two models of inward-facing conformations.

We appreciate the reviewers for bring up this possibility. In the second paragraph of the subsection “Molecular Dynamics Simulations”, we discuss why we favor generating a model of the inward facing state of LaINDY using our biased MD-based approach over using a homology modeling-based approach applied to the inverted repeats of LaINDY. For the same reasons, we favor using our biased MD-based approach over a homology modeling-based approach applied to the inward facing state of VcINDY. Namely, our model is generated in the presence of a membrane environment, and confirmation of its stability in this environment via MD is built in to the approach. This approach also enables the transporter dynamics to be assessed before, after, and during the transition between states.

It would have been possible to use information from the inward facing state of VcINDY, rather than information from LaINDY’s inverted repeats, to generate inward facing targets for the biased MD simulations. Within the collective variable space used in our approach, the targets suggested by LaINDY’s inverted repeats are -4.16 Å for *z* and -19.5° for *θ*, and the targets suggested by VcINDY’s inward facing state are -8.34 Å for *z* and -22.1° for *θ*. Both sets of targets are reasonable, but we ultimately chose to use the information provided by the inverted repeats because LaINDY and VcINDY, while homologous, are different proteins. The stability of our inward facing LaINDY model at the *z* and *θ* targets (Figure 6—figure supplement 1C and D) and the accessibility of bound succinate from the cytosol (Figure 6—figure supplement 1J) demonstrate the quality of the inverted repeat-based model.

In our manuscript, this issue is now addressed in the newly added third paragraph of the subsection “Molecular Dynamics Simulations”.

4) Please discuss whether you have you spotted potential metastable intermediates or larger barriers of the transporter on the way between the two extreme states.

A free energy profile is used to identify free energy barriers and potential metastable intermediate states along a transition, but the biased MD simulations we have performed do not generate such a profile. From our simulations, we are able to generate the work profiles required to induce the transitions. The presence of features in such profiles is sometimes used to propose the potential existence of barriers in a free energy profile, but there are no such features present in the work profiles associated with our induced transitions. While the biased MD simulations lay the groundwork needed to apply the advanced array of MD-based techniques needed to obtain a free energy profile, these techniques are quite computationally expensive and are beyond the scope of this work.

In our manuscript, this is now addressed in the newly added last paragraph of the subsection “Molecular Dynamics Simulations”.

5) The use of biased MD is justified, as the time scale of the transport is way beyond what can be currently achieved. Nevertheless, the authors should remind the readers that the biases and restraints imposed in the simulations might have biased the results toward the proposed transport routes.

In our biased MD simulations, the pathway taken through the collective variable space has not been optimized, and we take care in this work not to perform analyses that rely on this pathway. Rather, we restrict our analyses to the global structural differences between the end states and the internal dynamics of the transport domains, which this work suggests move as rigid bodies relative to the scaffold domain. The internal structural dynamics of the transport domains could have been affected by the restraints placed on succinate in simulation trials A_2_ and B_2_, but the consistency of the data from these trials with the data from trial B_1_, in which no restraints were placed on succinate, suggest that these restraints did not affect the transport domains’ internal dynamics.

In our manuscript, this point is now addressed in the newly added last paragraph of the subsection “Molecular Dynamics Simulations”.

6) It would be of interest to see whether the conformation of LaINDY will change if the mutations are made to either/both of the Arg159/Glu146 and His392/His401 pairs in the equivalent Na-sites.

We absolutely agree that an interrogation of the residues which occupy the Na1 and Na2 sites would be interesting. We are very interested in pursuing such a study in the future. However, such a set of experiments is beyond the scope of the current study.

7) Figure 1—figure supplement 2. Clones #76, 78 and 83 are said to be identical, but have different A450 readings for Na^+^ and Na^+^ + succinate states in Figure 1—figure supplement 2. If they are identical, how this is possible, especially as #78 has a higher value for T5 than T6, whereas #76 and #83 have the opposite pattern?

We than the reviewers for pointing out these potentially confusing results. This type of ELISA screening is only semi-quantitative and primarily used to identify binding-positive clones. In this particular case, the signals were weaker than typical. Consequently, the signals for these 76, 78 and 83 phage samples show the level of variation of this assay under the conditions used. To minimize potential confusion, we have updated a sentence in the figure legend, "Fab84 (indicated by a *), which has a higher affinity for the succinate bound state of VcINDY, was used for structure determination." to "Fab84 (indicated by a *), which has the highest signal for the succinate bound state of VcINDY, was used for structure determination."

8) Subsection “Structure determination of multiple states” and Figure 2. The authors convincingly showed that succinate can be transported by LaINDY, but they observe little/no succinate-induced stabilisation. Is 10 mM substrate not sufficient to stabilise the protein? Or do the authors anticipate a low binding affinity for succinate under these conditions?

We appreciate the reviewer’s attention to detail in the differences between substrate transport and small molecule binding and thermostabilization. We have previously observed concentration dependent stabilization effects in other transporters. At certain concentrations, a substrate can even thermodynamically destabilize the protein. There are a number of possible mechanisms, including binding affinity and the conformation of the protein in detergent solubilization, but a detailed examination is beyond the scope of this study. Therefore, here we simply use the thermostability assay to identify small molecules which stabilize LaINDY for structural studies.

We selected succinate for the transport assays for two reasons. Succinate is often the most common dicarboxylate in the cytoplasm. Its export has previously been shown to be central to the physiological role of other DASS exchangers in bacteria, and therefore we hypothesized it would be a substrate of LaINDY. Also, as the simplest C4 dicarboxylate, we expected succinate would bind and be transported by LaINDY, even if not the optimal substrate.

9) Subsection “Structure determination of multiple states”. Please explain the use of terephthalate instead of succinate in this X-ray structure. Is there a rationale for this?

Terephthalate is a synthetic dicarboxylate we hypothesized would act as a substrate surrogate for VcINDY, with two carboxylate moieties at approximately the same distance as those in succinate. We expected its larger planar benzene ring would allow us to unambiguously resolve the molecule within the VcINDY binding site. This is particularly relevant as the VcINDY X-ray structures to-date are all of modest resolution. These hypotheses of TTP binding and electron scattering proved correct in the determined structure. This has been explicitly stated in the subsection “Structure determination of multiple states”.

10) The authors identify the "LaINDY handle", but do not mention it again. Do the authors think it is mechanistically significant?

While this structural feature is visually distinct in the LaINDY structure, there is no evidence it has any significance to the transporter’s function. We have therefore removed the “LaINDY handle” name.

11) Subsection “LaINDY has Na^+^ surrogate side-chains near the substrate binding site”. The authors only mention charged residue surrogates for Na1 and Na2, but VcINDY requires 3 Na^+^ ions. Does LaINDY offer up any clues as to the location of the 3rd Na^+^ site? There is 10 mM Na^+^ in the reaction buffer in the transport assay. Is it certain that LaINDY (or other DASS exchangers) is Na^+^-independent?

The reviewers made an important point. Examining the LaINDY maps, we do not see any unmodeled densities which could be assigned as sodium with confidence. Therefore, they do not offer any guidance to the Na^+^ sites in VcINDY. As noted previously in point 1, we have not examined the sodium dependence of LaINDY. We have clarified these points in the second paragraph of the Discussion.

12) Subsection “Substrate release from VcINDY causes significant structural changes”. The authors compare the previous detergent-solubilised VcINDY structures that have Na^+^ and succinate bound to the amphipol solubilised VcINDY with Na^+^ only bound and highlight prominent changes to periplasmic surface. Are the authors suggesting that these changes are due to the substrate bound state of the protein? If so, is it not possible that these changes are merely due to a detergent vs. amphipol effect?

We thank the reviewers for pointing out this important technical issue. However, it is unlikely that the differences in the new VcINDY C_i_-Na^+^ structures, versus the previously described C_i_-Na^+^-S structures, are the results of amphipol or detergent effects. Notably, we determined two VcINDY C_i_-Na^+^ structures, in amphipol and in lipid nanodiscs. The structural features between these new cryo-EM structures are generally very consistent. Therefore, the C_i_-Na^+^ structural features are not likely the result of amphipol effects. This has been clarified in the subsection “Substrate release from VcINDY causes significant structural changes”.

For C_i_-Na^+^-S state, all the X-ray structures of VcINDY determined in detergent are essentially the same. However, it is fair to note that all of these structures were determined in the same detergent and crystallized in similar conditions with similar lattice contacts.

Therefore, while we cannot absolutely rule-out artifacts in the previous detergent-based X-ray structures, we are confident in the consistent structural differences between the C_i_-Na^+^ and C_i_-Na^+^-S states.